# The human origin recognition complex is essential for pre-RC assembly, mitosis, and maintenance of nuclear structure

Hsiang-Chen Chou[1,2†], Kuhulika Bhalla[1†], Osama EL Demerdesh[1], Olaf Klingbeil[1], Kaarina Hanington[1], Sergey Aganezov[3], Peter Andrews[1], Habeeb Alsudani[1], Kenneth Chang[1], Christopher R Vakoc[1], Michael C Schatz[3], W Richard McCombie[1], Bruce Stillman[1]*

[1]Cold Spring Harbor Laboratory, Cold Spring Harbor, United States; [2]Graduate Program in Molecular and Cellular Biology, Stony Brook University, Stony Brook, United States; [3]Department of Computer Science, Whiting School of Engineering, Johns Hopkins University, Baltimore, United States

**Abstract** The origin recognition complex (ORC) cooperates with CDC6, MCM2-7, and CDT1 to form pre-RC complexes at origins of DNA replication. Here, using tiling-sgRNA CRISPR screens, we report that each subunit of ORC and CDC6 is essential in human cells. Using an auxin-inducible degradation system, we created stable cell lines capable of ablating ORC2 rapidly, revealing multiple cell division cycle phenotypes. The primary defects in the absence of ORC2 were cells encountering difficulty in initiating DNA replication or progressing through the cell division cycle due to reduced MCM2-7 loading onto chromatin in G1 phase. The nuclei of ORC2-deficient cells were also large, with decompacted heterochromatin. Some ORC2-deficient cells that completed DNA replication entered into, but never exited mitosis. ORC1 knockout cells also demonstrated extremely slow cell proliferation and abnormal cell and nuclear morphology. Thus, ORC proteins and CDC6 are indispensable for normal cellular proliferation and contribute to nuclear organization.

**\*For correspondence:**
stillman@cshl.edu

[†]These authors contributed equally to this work

## Introduction

Cell division requires the entire genome to be duplicated once and only once during S-phase of the cell cycle, followed by segregation of the sister chromatids into two daughter cells. To ensure complete and correct duplication of genomes, the initiation of DNA replication is highly regulated and begins with the assembly of a pre-Replication Complex (pre-RC) at origins of DNA replication throughout the genome (*Bell and Labib, 2016*). Among eukaryotes, *Saccharomyces cerevisiae* is the best characterized system, from which individual proteins involved in DNA replication have been identified and studied extensively, including functional reconstitution of the entire pre-RC assembly and the regulated initiation of DNA replication from these pre-RCs with purified proteins (*Evrin et al., 2009*; *Remus et al., 2009*; *Yeeles et al., 2015*). In *S. cerevisiae*, pre-RC assembly begins with the origin recognition complex (ORC), comprising Orc1-6 subunits, binding to each potential DNA replication origin (*Bell et al., 1993*; *Bell and Labib, 2016*; *Bell and Stillman, 1992*; *Gibson et al., 2006*). Chromatin-bound ORC then provides a platform for the assembly and recruitment of other pre-RC proteins. Cdc6 binds to ORC, followed by the binding of Cdt1-Mcm2-7 to form head-to-head Mcm2-7 double hexamers to complete the formation of the pre-RC (*Araki, 2011*; *Bell and Labib, 2016*; *Bleichert et al., 2017*; *Evrin et al., 2009*; *Heller et al., 2011*; *Remus et al., 2009*). The Mcm2-7 double hexamer helicase precursor complex remains bound to DNA in an inactive state until it is activated by additional proteins and protein kinases (*Bell and Labib, 2016*). During S phase, cyclin-dependent protein kinase (CDK) and the Cdc7-Dbf4-dependent protein kinase

(DDK), Sld2, Mcm10, Dpb11, Sld3/7, DNA polymerase ε, Cdc45 and the GINS complex are recruited to activate MCM2-7 helicase (*Araki, 2016*; *Araki et al., 1995*; *Bell and Labib, 2016*; *Kamimura et al., 2001*; *Kamimura et al., 1998*; *Takayama et al., 2003*; *Yeeles et al., 2015*). The functional helicase consists of Cdc45-Mcm2-7-GINS (CMG) and when activated it unwinds the DNA in a bidirectional and temporally regulated manner from each origin (*Bleichert et al., 2017*).

In all eukaryotes, including *S. cerevisiae* and human cells, the ORC1-5 subunits contain a AAA+ or a AAA+-like domain and a winged-helix domain (WHD) (*Bleichert et al., 2017*; *Chen et al., 2008*; *Jaremko et al., 2020*; *Li et al., 2018*; *Ocaña-Pallarès et al., 2020*; *Tocilj et al., 2017*). In yeast, Orc1-6 remains as a stable complex bound to the chromatin throughout the cell division cycle (*Aparicio et al., 1997*; *DePamphilis, 2003*; *Weinreich et al., 1999*). ORC binds to A and B1 DNA sequence elements within the autonomously replicating sequence (ARS), which contains a conserved ARS consensus sequence (ACS) (*Bell and Labib, 2016*; *Bell and Stillman, 1992*; *Celniker et al., 1984*; *Deshpande and Newlon, 1992*; *Marahrens and Stillman, 1992*; *Rao and Stillman, 1995*; *Rowley et al., 1995*). On the other hand, in human cells, there is no apparent sequence-specific binding of ORC to DNA, and the binding of ORC to chromosomes is dynamic (*Vashee et al., 2001*). ORC subunits do, however, localize to specific sites within the chromosome, most likely via interactions with modified histones or other chromatin-interacting proteins (*Higa et al., 2017*; *Hossain and Stillman, 2016*; *Kuo et al., 2012*; *Long et al., 2019*; *Miotto et al., 2016*; *Tatsumi et al., 2008*). One or more of the human ORC subunits dissociate from the complex soon after the pre-RC is formed. For example, in human cells ORC1 is ubiquitinated by the SCF^skp2 ubiquitin ligase and is degraded during the G1-S transition and early S phase, and then re-appears as cells enter mitosis (*Kara et al., 2015*; *Kreitz et al., 2001*; *Méndez et al., 2002*; *Ohta et al., 2003*). In human cells, ORC1 is the first ORC subunit to bind to mitotic chromosomes and is inherited into the daughter cells where it recruits other ORC subunits and CDC6 to form new pre-RCs (*Kara et al., 2015*; *Okuno et al., 2001*).

ORC is a conserved complex in eukaryotes, and it is essential for DNA replication in *S. cerevisiae*, *S. pombe*, Xenopus, *and Drosophila*, since mutation or depletion of ORC prevents CDC6 binding and MCM loading onto DNA (*Aparicio et al., 1997*; *Chuang et al., 2002*; *Pak et al., 1997*; *Pflumm and Botchan, 2001*; *Romanowski et al., 1996*; *Speck et al., 2005*). Besides its function in the initiation of DNA replication, ORC protein subunits also have other important roles that vary with species. In budding yeast, Orc1 directly interacts with silencing regulator Sir1 at the silent mating type loci to mediate transcriptional gene silencing and maintain heterochromatin (*Bell et al., 1993*; *Foss et al., 1993*; *Fox et al., 1995*; *Hou et al., 2005*; *Triolo and Sternglanz, 1996*). ORC1 also plays a role in transcriptional gene silencing of the human *CCNE1* locus in human cells (*Hossain and Stillman, 2016*). ORC also interacts with heterochromatin protein HP1 and is required for maintenance of heterochromatin (*Pak et al., 1997*; *Pflumm and Botchan, 2001*; *Prasanth et al., 2010*; *Prasanth et al., 2004*; *Shen et al., 2012*).

ORC2 depletion after pre-RC assembly resulted in spindle and DNA damage checkpoint activation, and impaired sister-chromatid cohesion (*Shimada and Gasser, 2007*). In *Drosophila*, Orc2 mutants showed reduced S phase cells, increased number of mitotic cells with abnormally condensed chromosomes and chromosome alignment defects, and more importantly, those mutants could not survive at late larval stage (*Loupart et al., 2000*; *Pflumm and Botchan, 2001*). In humans, mutations in ORC1, ORC4, ORC6, CDT1, and CDC6 are detected in Meier-Gorlin syndrome (MGS) patients (*Bicknell et al., 2011b*; *Bicknell et al., 2011a*; *Guernsey et al., 2011*; *Hossain and Stillman, 2012*; *de Munnik et al., 2015*). ORC1 and ORC2 localize to centrosomes and ORC1 regulates the re-duplication of the centriole (*Hemerly et al., 2009*; *Prasanth et al., 2004*). ORC also localizes to telomeres via the TRF2 shelterin protein (*Deng et al., 2009*; *Tatsumi et al., 2008*). It was also shown that siRNA knockdown or CRISPR/Cas9 knockout of ORC1 resulted in loss of MCM2-7 from chromatin, abnormal duplication of centrioles, and a change in cell cycle stage distribution (*Hemerly et al., 2009*; *Kara et al., 2015*; *McKinley and Cheeseman, 2017*). ORC1, ORC2, ORC3, and ORC5 associate with heterochromatin, and depletion of ORC subunits disrupt localization of heterochromatin and also causes abnormal heterochromatin decondensation in cells (*Giri et al., 2016*; *Prasanth et al., 2010*; *Prasanth et al., 2004*). ORC2 and ORC3 also specifically localize to centromeric heterochromatin during late S phase, G2 and mitosis and removal of these proteins causes decondensation of centromeric α-satellite (*Craig et al., 2003*; *Prasanth et al., 2010*; *Prasanth et al., 2004*).

There is an emerging debate, however, about the essential nature of ORC in human cells (*Bell, 2017*). ORC is overexpressed in numerous cancerous cell lines (*McNairn and Gilbert, 2005*) and HCT116 colorectal cancer cells can survive with only 10% of the ORC2 protein level (*Dhar et al., 2001*). More importantly, it was reported that HCT116 $p53^{-/-}$ ($TP53^{-/-}$, but we henceforth use $p53^{-/-}$) cells in which expression of either ORC1 or ORC2 subunit was eliminated using CRISPR-Cas9-mediated gene ablation could still proliferate (*Shibata et al., 2016*). Here, we developed a genetic method to address the function of the pre-RC proteins ORC and CDC6, particularly focusing on the ORC1 and ORC2 subunits. We demonstrate that ORC proteins are essential for normal cell proliferation and survival of human cells. Moreover, ORC1 or ORC2-depleted cells showed multiple defects in progression through cell division cycle, including DNA replication and mitosis, as well as defects in nuclear structure.

## Results

### ORC1-6 and CDC6 are essential for cell survival

To address the issue of essentiality and to identify functional domains within the ORC and CDC6 proteins, we used unbiased tiling-sgRNA CRISPR negative selection screens. Evaluation of CRISPR knock-out (CRISPR-KO) strategies have shown that targeting regions within protein domains typically show significantly higher degree of negative selection phenotypes (*He et al., 2019*; *Hsu et al., 2018*; *Montalbano et al., 2017*; *Munoz et al., 2016*; *Shi et al., 2015*; *Wang et al., 2019*). This is because both frameshift and more crucially in-frame mutations within functionally active regions of a protein result in genetic nulls (*Munoz et al., 2016*; *Shi et al., 2015*). Off-target effects notwithstanding, targeting known domains that contribute to protein function have informed the design of pooled whole-genome CRISPR screens like GeCKO and Avana. Similarly, applying CRISPR-KO strategies to individual proteins requires selection of a single or a few sgRNAs that target known functional domains and have a considerably low off-target score. However, at least in the case of ORC, CRISPR-based ablation of individual subunits of the complex have reported different phenotypic outcomes. For example, in DepMap, the database that summarizes results from whole-genome CRISPR screens (GeCKO 19Q1 and Avana 20Q2 libraries) (*Meyers et al., 2017*; *Tsherniak et al., 2017*), *ORC1* is listed as a common essential gene while *ORC2* is listed as non-essential in tested cell types (*Figure 1—figure supplement 2a*). Other members of the pre-RC proteins – *ORC3*, *ORC4*, *CDC6*, *MCM2-7*, and *CDT1* show variability between GeCKO and Avana screens in being described as common essential or not. In 2015, *ORC1*, *ORC4*, and *MCM4* were reported as essential in murine cells by using guides that targeted the AAA+ or WH domains of the proteins (*Shi et al., 2015*). Subsequently, in human colorectal carcinoma cell line HCT116, *ORC1*, and *ORC2* were reported to be non-essential for cell proliferation (*Shibata et al., 2016*).

The rationale for using a pooled tiling-sgRNA CRISPR screen approach was – (a) since essential protein domains correlate with higher negative selection phenotype, we hypothesized that analyzing the effect of every possible guide RNA target site in the ORF might uncover new functional regions, and (b) a high-throughput screen of this nature would provide incontrovertible evidence about the essentiality of ORC and CDC6 proteins, at least in the cell lines tested. A recent study validated this approach by analyzing tiling-sgRNA data from Munoz et. al. and found that up to 17.7% of the regions that displayed a CRISPR knockout hyper-sensitive (CKHS) phenotype, did not overlap with previously annotated domain or known function (*He et al., 2019*; *Munoz et al., 2016*). Thus, guide RNAs targeting every possible PAM sequence 5′-NGG-3′ (*Streptococcus pyogenes* Cas9) across each exon of *ORC1-6* and *CDC6* were designed and synthesized. Pooled CRISPR libraries also included control guide RNAs targeting either known core essential genes such as *CDK1*, *PCNA* etc. as positive controls, or those targeting non-essential gene loci or no loci at all as negative controls (*Miles et al., 2016*). The total library comprised 882 guides targeting *ORC1-6* and *CDC6*, 1602 negative controls (Used in GeCKO V2 library - 'NeGeCKO' (*Sanjana et al., 2014*), negative controls used in The Sabatini/Lander CRISPR pooled library (*Park et al., 2017*), *Rosa26*, CSHL in-house negatives (*Lu et al., 2018*; *Tarumoto et al., 2018*) and 43 positive controls; with a median of three pre-validated guides targeting known essential genes *CDK1*, *CDK9*, *RPL9*, *PCNA* etc; *Supplementary file 1*_guides). Parallel screens were done in the colorectal cancer derived HCT116-Cas9 cells and human diploid RPE-1-Cas9 cells and the relative depletions of guide RNAs in the cell

populations between Day 3 and Day 21 were compared using the guide read counts generated by Illumina based next-generation sequencing (n = 2 for HCT116, n = 1 for RPE-1) and the data was analyzed with MAGeCK (*Li et al., 2014*). The screens performed well as shown by the consistent log fold change (LFC) pattern of depletion or relative enrichment of positive and negative controls, respectively – although the absolute values and the range of LFCs were cell-line specific. The LFC threshold of 'essentiality' for each cell line was set at the value at which a guide RNA was depleted more than every negative control as well as $\geq$ to the median depletion of guides targeting each positive control (*Figure 1—figure supplement 1b–d*, red line). In HCT116, LFC $\leq -1$ and LFC $\leq -5$ in RPE-1 were found to be the cut-off for log fold depletion.

The results showed significant depletion of guide RNAs that target regions within structurally defined domains (*Figure 1a–b and f*, *Figure 2a–c*, *Figure 2—figure supplements 3a–c, g*, *4a–c, g, h–j, n*, *5a–c, g, h–j and n*). To visualize the tiling-sgRNA data relative to amino acid conservation and intrinsic disorder, we used NCBI RefSeq coding sequences (NP_004144.2, NP_006181.1, NP_862820.1, NP_859525.1, NP_002544.1, NP_055136.1, NP_001245.1) for three analyses - (1) FrPred (*Adamczak et al., 2004*; *Fischer et al., 2008*) (https://toolkit.tuebingen.mpg.de/frpred) server that calculates a conservation score based on amino acid variability as well as the probability of it being a functional ligand binding or catalytic site at each amino acid position of the input sequence (*Figure 1c*, *Figure 2—figure supplements 1b*, *3d*, *4d, k*, *5d and k*); (2) Consurf (*Ashkenazy et al., 2016*) (https://consurf.tau.ac.il/) server which analyses the probability of structural and functional conservation despite amino acid variability for any given position of input sequence (*Figure 1d*, *Figure 2—figure supplements 1c*, *3e*, *4e, l*, *5e and l*). We ran these analyses with default parameters except for the number of species to include. In one analysis, we chose 50 representative homologues with maximum and minimum percent identity set to 95 and 50 across species. In the other, we increased the species to 150 and set max. and min. percent identity to 95 and 35 to compare a larger evolutionary subset. In both analyses, the UNIREF90 database was used, which consists of cluster sequences that have at least 90% sequence identity with each other into a single UniRef entry, thus increasing the representative diversity of species considered in the output. And lastly, (3) Disopred tool (*Buchan and Jones, 2019*; *Jones and Cozzetto, 2015*) (http://bioinf.cs.ucl.ac.uk/psipred/) that scores for intrinsically disordered regions (IDRs) that are usually not well conserved yet found to be functionally essential in many proteins (*Figure 1e*, *Figure 2—figure supplements 1d*, *3f*, *4f, m*, *5f and m*).

When the DepMap CRISPR Achilles (Avana 20Q2 library) dataset was compared to a combined RNAi dataset of cell lines, it indicated that using the CRISPR method, with a gene effect score of less than $-0.5$, *ORC1* classified as common essential in > 90% of the cell lines, while with RNAi datasets with that same cut-off, it classified as essential in only about 45% of the same cell lines (*Figure 1—figure supplement 2b–c*). It is evident that the method of choice did have a bearing on the phenotypic outcome of the knock-down. The study by *Shibata et al., 2016* that found *ORC1* and *ORC2* to be non-essential also used CRISPR editing as the method of knock-down, but also performed long-term selection for cell proliferation to obtain *ORC1*$^{-/-}$ of *ORC2*$^{-/-}$ cells. We therefore determined if our screen had guide RNAs that were used in either of the DepMap dataset or used in the directed study (*Shibata et al., 2016*). For *ORC1* and *ORC2* sgRNAs that were used in DepMap datasets, there was a variation in their phenotype as measured by LFC values, with some guides classifying *ORC1* and *ORC2* as essential and others not (*Figure 2—figure supplement 2a–c*). Of note is the fact that the guide used to target *ORC2* in the *Shibata et al., 2016* study showed activity very close to the cut-off in HCT116 cells and scored as non-essential in RPE-1. It is important to note that when a guide targeting a relatively non-essential region allows for the cells to proliferate, no conclusion can be made about the protein being essential. The *Shibata et al., 2016* study used that single guide to insert a gene encoding blasticidin resistance and a poly A cassette into the locus, with the aim of disrupting transcription, while our single-guide-per-locus type of screen did not introduce such large insertions. We find that *ORC1-6* and *CDC6* are all essential in both cell lines tested, and that the dynamic range of depletion of these proteins was greater in diploid RPE-1 cells (*Figure 2—figure supplement 7a*). Comparison of the results from our screens and the published DepMap, especially about *ORC2*, suggest that using too few guides to target proteins can lead to artifactual observations both in terms of essentiality or non-essentiality, and that overall, the gene-effect is influenced by the combination of the choice of guide RNA and the cell line studied.

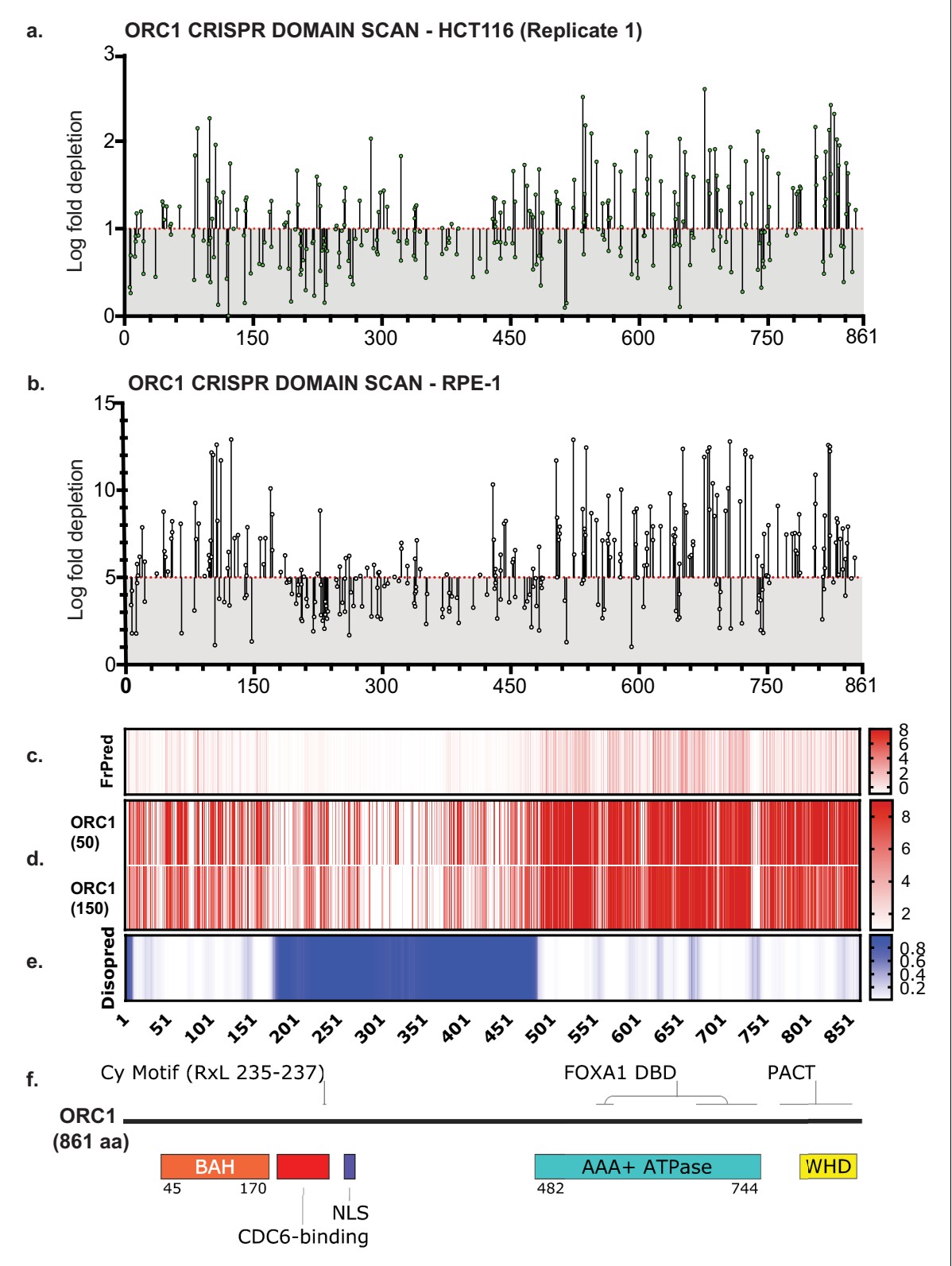

**Figure 1.** ORC1 is essential to HCT116 and RPE-1 cell lines. (**a**) Tiling-sgRNA map ORC1 (replicate 1) in HCT116. Mapped as Log fold depletion (inverted LFC scale) of guide RNAs as calculated by MAGeCK (**Li et al., 2014**) on y axis vs the amino acid(s) disrupted by that guide RNA on the x axis. Effect of guide RNA is interpreted as essential if its depletion is more than one log fold (red dotted line). Data mapped on 1 Log fold depletion pseudo-axis for clarity (See **Figure 1—source data 1**). (**b**) Tiling-sgRNA map of ORC1 in RPE-1. Effect of guide RNA is interpreted as essential if its

*Figure 1 continued*

depletion is more than five log fold (red dotted line). Data mapped on 5 Log fold depletion pseudo-axis for clarity (See *Figure 1—source data 2*). (**c**) FrPred (https://toolkit.tuebingen.mpg.de/frpred) of hORC1 (NP_004144.2) shown as gradient heat map of conservation score vs amino acid position. (**d**) Consurf (https://consurf.tau.ac.il/) of hORC1 – (upper) ORC1 (50) subset (50 HMMER Homologues collected from UNIREF90 database, Max sequence identity = 95%, Min sequence identity 50, Other parameters = default), and (lower) ORC1 (150) subset (150 HMMER Homologues collected from UNIREF90 database, Max sequence identity = 95%, Min sequence identity 35, Other parameters = default). Data represented as heat map of Conservation scores of each amino acid position. (**e**) Disopred (http://bioinf.cs.ucl.ac.uk/psipred/) plot of hORC1 – heat map representing amino acids within intrinsically disordered regions of the protein. (**f**) Schematic of domain architecture of ORC1.

The online version of this article includes the following source data and figure supplement(s) for figure 1:

**Source data 1.** Numerical data table for ORC1 tiling sgRNA CRISPR screen log fold depletion in *Figure 1a*.
**Source data 2.** Numerical data table for ORC1 tiling sgRNA CRISPR screen log fold depletion in *Figure 1b*.
**Figure supplement 1.** Tiling-sgRNA CRISPR screen data and controls.
**Figure supplement 2.** DepMap analyses of *ORC1* data.

When we evaluate the distribution of LFC for all sgRNAs, we saw a negative correlation with annotated domains (AD) – that is sgRNAs targeting AD regions showed significantly higher depletion compared to those targeting non-annotated domain (NAD) regions (*Figure 2—figure supplements 6a* and *7a*). This finding is consistent with the only previous study that used tiling-sgRNA negative CRISPR screens (*Munoz et al., 2016*). Moreover, we see a high correlation between our datasets from both HCT116 replicates ($r \approx 0.7$) as well as between HCT116 and RPE-1 ($r \approx 0.47$), which is also consistent with the previous study that compared three cell lines (*Figure 2—figure supplement 7b*). For each ORC subunit (and CDC6), $\geq$50% of designed sgRNAs target annotated domain (AD) regions (*Figure 2—figure supplement 6a*), with the exception of *ORC4*, which is entirely structured, and thus none of the sgRNAs target NAD regions. For all other genes, the fraction of NAD targeting sgRNAs that scored as essential, as well as their locations within these regions were comparable between cell lines rather than between proteins (*Figure 2—figure supplement 6b–h*). This suggests there are functional modules within the NADs and indeed, at least in some cases, we found that sgRNA dropouts in our screens agree with recent functional studies about these regions. For example, in *ORC1*, in HCT116 we saw depletion of sgRNAs targeting regions between 300–450 aa. In humans and *Drosophila*, this N-terminal region is an unstructured IDR, required for ATP-independent chromatin recruitment of ORC and drives protein phase-separation with DNA in vitro (*Bleichert et al., 2018*; *Hossain et al., 2019*; *Parker et al., 2019*). In yeast, this region of Orc1 has been shown to be critical for interacting with ARS DNA as well, but is completely structured unlike metazoan ORC1 (*Hu et al., 2020*; *Kawakami et al., 2015*; *Li et al., 2018*). Another example is ORC6 in which the N-terminus harbors a structured TFIIB domain, but we noticed sgRNA depletions in the remaining regions, especially the C-terminal extremity, which is a septin-binding-region essential for cytokinesis and mutations in a conserved C-terminal motif have also been linked to Meier-Gorlin Syndrome (*Akhmetova et al., 2015*; *Balasov et al., 2020*; *Balasov et al., 2015*; *Balasov et al., 2007*; *Bicknell et al., 2011a*; *Prasanth et al., 2002*; *Xu et al., 2020*).

Limitations like variability of sgRNA efficiency, target gene copy number, *p53* status of the cell, post-translational modifications, local structure of the gene locus can confound analyses from high-throughput tiling-sgRNA CRISPR screening (*Haapaniemi et al., 2018*; *Munoz et al., 2016*). To determine similarities or differences in 'regions of essentiality' in ORC and CDC6 between HCT116 and RPE-1 cell lines, we used the computational tool Protiler, developed specifically for tiling-sgRNA CRISPR screens (*He et al., 2019*). This tool takes into account local outliers due to inactive sgRNA or additive effects, and maps sgRNA depletion signals to amino acids of the target proteins to identify functionally essential regions based on their CRISPR-knockout hyper-sensitivity (CKHS). MAGeCK LFC values for combined HCT116 replicates and RPE-1 were put through this pipeline at two different -t2/–thresholds, 0.25 and 0.5, which detect changing points using the TGUH method described in this study. Almost all annotated domains or their boundaries overlapped with CKHS regions (*Figure 2—figure supplements 8–13*). In addition, in ORC1, ORC2, ORC3, ORC6, and CDC6 NAD regions were also determined to be CKHS. Some of the newly identified CKHS regions are in agreement with studies that have found that the N-terminal IDR region of ORC1, ORC2, and CDC6, and a small C-terminal region of ORC6, to be functionally important and therefore indispensable

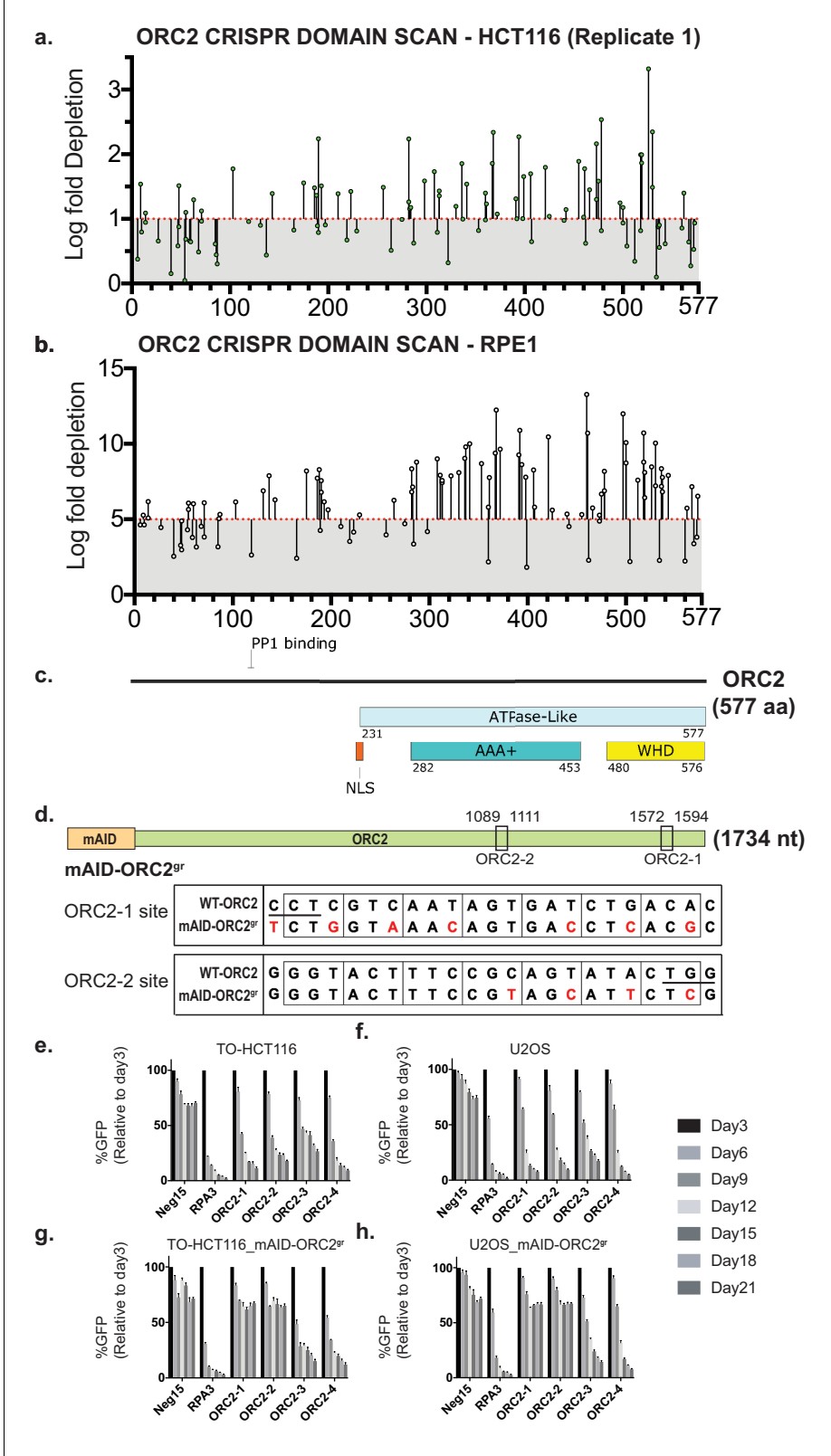

**Figure 2.** *ORC2* is essential in HCT116 and RPE-1 by both tiling-sgRNA and single guide CRISPR knock-down. (a) Tiling-sgRNA map of *ORC2* (replicate 1) in HCT116. Mapped as Log fold depletion (inverted LFC scale) of guide RNAs as calculated by MAGeCK (*Li et al., 2014*) on y axis vs the amino acid (s) disrupted by that guide RNA on the x axis. Effect of guide RNA is interpreted as essential if its depletion is more than one log fold (red dotted line). Data mapped on one log fold depletion pseudo-axis for clarity (See *Figure 2—source data 1*). (b) Tiling-sgRNA map of *ORC2* for RPE-1. Data mapped

*Figure 2 continued on next page*

*Figure 2 continued*

on 5 Log fold depletion pseudo-axis for clarity (See *Figure 2—source data 2*). (c) Schematic of ORC2 protein showing annotated structural or functional domains. (d) The top panel is the schematic of the mAID degron fused to *ORC2* transgene at the N-terminus, and the two black rectangles indicate ORC2-1 and ORC2-2 sgRNAs targeting regions. The numbers represent nucleotide positions in the *ORC2* cDNA. The lower two panels show the silent mutations (in red) around the sgRNA target sites introduced into mAID-ORC2$^{gr}$ compared to wild-type ORC2. Protospacer-adjacent motif (PAM) site is underlined in the wild-type sequence. (e–h) Negative-selection time course assay that plots the percentage of GFP-positive cells over time following transduction with the indicated sgRNAs. Experiments were performed in (e) TO-HCT116, (f) U2OS, (g) TO-HCT116_mAID-ORC2$^{gr}$, and (h) U2OS_mAID-ORC2$^{gr}$ cell lines. The GFP-positive percentages were normalized to the Day3 measurement. n = 3. Error bars, mean ± SD.

The online version of this article includes the following source data and figure supplement(s) for figure 2:

**Source data 1.** Numerical data table for ORC2 tiling sgRNA CRISPR screen log fold depletion in *Figure 2a*.
**Source data 2.** Numerical data table for ORC1 tiling sgRNA CRISPR screen log fold depletion in *Figure 2b*.
**Figure supplement 1.** (a) Tiling-sgRNA map of *ORC2* (replicate 2) in HCT116.
**Figure supplement 2.** Table of guide RNAs.
**Figure supplement 3.** Tiling-sgRNA CRISPR screen data contd.
**Figure supplement 4.** Tiling-sgRNA CRISPR screen data contd.
**Figure supplement 5.** Tiling-sgRNA CRISPR screen data contd.
**Figure supplement 6.** Analysis of ORC1-6, CDC6 tiling-sgRNA CRISPR screens.
**Figure supplement 7.** Guide RNAs targeting annotated domains show a higher negative selection phenotype.
**Figure supplement 8.** Identification of CKHS regions in *ORC1* using Protiler.
**Figure supplement 9.** Identification of CKHS regions in *ORC2* using Protiler.
**Figure supplement 10.** Identification of CKHS regions in *ORC3* using Protiler.
**Figure supplement 11.** Identification of CKHS regions in *ORC4* and *ORC5* using Protiler.
**Figure supplement 12.** Identification of CKHS regions in *ORC6* using Protiler.
**Figure supplement 13.** Identification of CKHS regions *CDC6* using Protiler.

(*Akhmetova et al., 2015*; *Balasov et al., 2007*; *Hossain et al., 2019*; *Lidonnici et al., 2004*; *Parker et al., 2019*; *Prasanth et al., 2002*; *Shen et al., 2012*).

At this point, we selected sgRNAs that fall within regions of AD and CKHS overlap to target *ORC2* and characterize the phenotype of such ablation (*Figure 2—figure supplement 2d*). We also received *ORC1* and *ORC2* deficient stable cell lines from the authors of the previous study (*Shibata et al., 2016*) for further analysis.

## Rapid ORC2 removal in cancer cells impedes cell growth and causes DNA damage

Knock-down of *ORC2* with an siRNA approach was a slow process that took at least 24–48 hr, however, using this method we have observed various defective phenotypes, including G1 arrest, S-phase defects, abnormally condensed chromosomes as well as defects in mitosis (*Prasanth et al., 2010*; *Prasanth et al., 2004*). These phenotypes can be outcomes of accumulated errors that happen during any phase of the cell cycle and thus it is hard to distinguish between primary and secondary phenotypes associated with the loss of ORC2. Therefore, we used CRISPR/Cas9 in combination with an auxin inducible degron (mAID) tagged ORC2 to construct cell lines in which endogenous ORC2 could be knocked out by CRISPR, and the complementing CRISPR-resistant mAID-ORC2 could then be rapidly removed from cells, allowing exploration of the importance of ORC2 at different stages of cell division cycle (*Natsume et al., 2016*; *Nishimura et al., 2009*). To mediate the endogenous ORC2 knockout, four sgRNAs, hereafter named ORC2-1, ORC2-2, ORC2-3, and ORC2-4, were selected for on-target single guide validation. For complementation, N-terminally tagged mAID-sgRNA resistant ORC2 (mAID-ORC2$^{gr}$) was constructed, and the *ORC2* cDNA was edited to harbor multiple mismatches based on two of the four sgRNAs, ORC2-1 and ORC2-2 (*Figure 2d*). To perform ORC2 depletion or genetic complementation, mAID-ORC2$^{gr}$ constructs were transduced into two cell lines, TO-HCT116 cell line, which expresses a doxycycline-inducible *Oryza sativa* (Asian rice) *TIR1* (OsTIR1) gene that encodes a plant auxin-binding receptor that interacts with the conserved E3 ubiquitin ligase SCF complex to degrade mAID-tagged proteins and the U2OS cell line. In the dropout CRISPR/Cas9 experiment, cells expressing a positive control RPA3 sgRNA and all four ORC2 sgRNAs, but not the negative control Neg15 sgRNA (CSHL in-house negatives, *Supplementary file 1*), showed depletion over 3 weeks of cell culture (*Figure 2e,f*). The effects of

ORC2-1 and ORC2-2 sgRNAs could be rescued by mAID-ORC2$^{gr}$ in both TO-HCT116_mAID-ORC2$^{gr}$ and U2OS_mAID-ORC2$^{gr}$ cell lines confirming target specificity (*Figure 2g,h*).

To acquire clonal cells to study ORC2 depletion phenotypes, TO-HCT116_mAID-ORC2$^{gr}$ cells were depleted of the endogenous *ORC2* gene with sgRNA ORC2-1 and single clones were isolated by flow sorting. Five cell lines, ORC2_H-1, ORC2_H-2, ORC2_H-3, ORC2_H-4, and ORC2_H-5, were obtained from two independent CRISPR/Cas9 knockout experiments done about 6 months apart. Sequencing of the target sites showed that the ORC2_H-1 and H-3 cell lines had heterozygous mutations at the sgRNA targeting site which led to premature stop codons downstream of the target site (*Figure 3—figure supplement 1a*). On the other hand, the H-2, H-4, and H-5 cell lines were homozygous with an identical two-nucleotide-deletion, creating a nonsense mutation at the sgRNA targeting site. Although the ORC2-1 sgRNA targets the C-terminus of ORC2, no truncated form of protein was detected by western blot. Our ORC2 rabbit polyclonal antibody was raised against the N-terminal half of ORC2 protein. The LTR-driven mAID-ORC2$^{gr}$ protein expressed at lower levels compared to endogenous ORC2 in TO-HCT116, RPE-1 and IMR90 cells, but was sufficient to complement the loss of endogenous ORC2 (*Figure 3a*).

For further analysis of the effects of auxin-induced ORC2 depletion in cells, ORC2_H-2, H-4, and H-5 cell lines were used because of their genetically identical deletions at the CRISPR cut-site and because they showed efficient depletion of mAID-ORC2$^{gr}$ after auxin treatment (*Figure 3*). We excluded off-target effects by confirming the ORC2_H-2 cell line was resistant to both ORC2-1 and ORC2-2 sgRNAs, but not to the ORC2-3 and ORC2-4 sgRNAs (*Figure 3—figure supplement 1b*). Compared with parental TO-HCT116 cell line, the human diploid cell RPE-1 expressed ~ 50% less ORC2, while IMR-90 cells expressed ~ 75% less (*Figure 3a*). The relative levels of ORC3 reflect the levels of ORC2 since they are known to form a cognate complex throughout the cell cycle (*Dhar et al., 2001*; *Jaremko et al., 2020*; *Vashee et al., 2001*). ORC2_H-2, H-4, and H-5 cells had no detectable endogenous ORC2, and ORC3 showed stoichiometrically comparable expression to mAID-ORC2$^{gr}$ levels (*Figure 3a*). In addition, ORC2_H-2 cells expressed mAID-ORC2$^{gr}$ at only about 5% that of endogenous ORC2 levels in TO-HCT116, while H-4 and H-5 cells expressed marginally more at about 10%. It is known that cancer cells can proliferate normally with 10% of the levels of ORC2 (*Dhar et al., 2001*).

Next, we compared the proliferation rates in these cell lines. In normal medium the ORC2_H-2, H-4, and H-5 cells grew slightly slower than the parental TO-HCT116 cells (*Figure 3b*). When doxycycline was added to induce OsTIR1 expression, proliferation of all cell lines decreased by similar rates, possibly either due to some toxicity to doxycycline or the expression of OsTIR1 protein itself (*Figure 3c*). It is important to note that auxin alone did not affect the proliferation rate of wild type TO-HCT116, H-4, and H-5 cells, but it reduced the proliferation rate of H-2 cells substantially (*Figure 3d*). This phenotype was probably caused by the leaky expression of Tet-OsTIR1 in ORC2_H-2 cells. Finally, when both doxycycline and auxin were added, all three ORC2 KO cell lines stopped proliferating entirely, whereas the parental TO-HCT116 cells continued to proliferate (*Figure 3e*).

Concomitant with the lack of cell proliferation, we observed altered cell cycle profiles after mAID-ORC2$^{gr}$ was depleted from these cells. Cells were treated with doxycycline and auxin to deplete mAID-ORC2$^{gr}$ for 0, 4, 24, and 50 hr. At the 50 hr time point, all three ORC2 KO cell lines had less cells progressing from G1 into S phase, and more cells accumulated in late S phase or the G2/M phase (*Figure 3f*, *Figure 3—figure supplement 2*). Cells with a 4C DNA peak (late S/G2/M phase) continued to incorporate EdU, suggesting that DNA replication was not complete, even though the bulk of the genome was duplicated. This phenotype was consistent with previous observations that cells treated with *ORC2* siRNA arrested in interphase (70%) or as rounded, mitotic-like cells (30%) (*Prasanth et al., 2004*).

To analyze whether the cell cycle arrest was due to checkpoint activation in response to DNA damage, cell extracts were prepared from doxycycline and auxin treated cells and analyzed by immunoblotting for various DNA damage markers. CHK1 is essential for the DNA damage response and the G2/M checkpoint arrest and is primarily phosphorylated by ATR, although phosphorylation by ATM has also been reported (*Gatei et al., 2003*; *Goto et al., 2019*; *Jackson et al., 2000*; *Liu et al., 2000*; *Wilsker et al., 2008*). ORC3 and mAID-ORC2$^{gr}$ proteins in ORC2_H-2, H-4, and H-5 cell lines were depleted following 4 hr of auxin treatment (*Figure 3g*). A nonspecific smaller band was detected but this band did not co-immunoprecipitate with ORC3 (*Figure 3—figure*

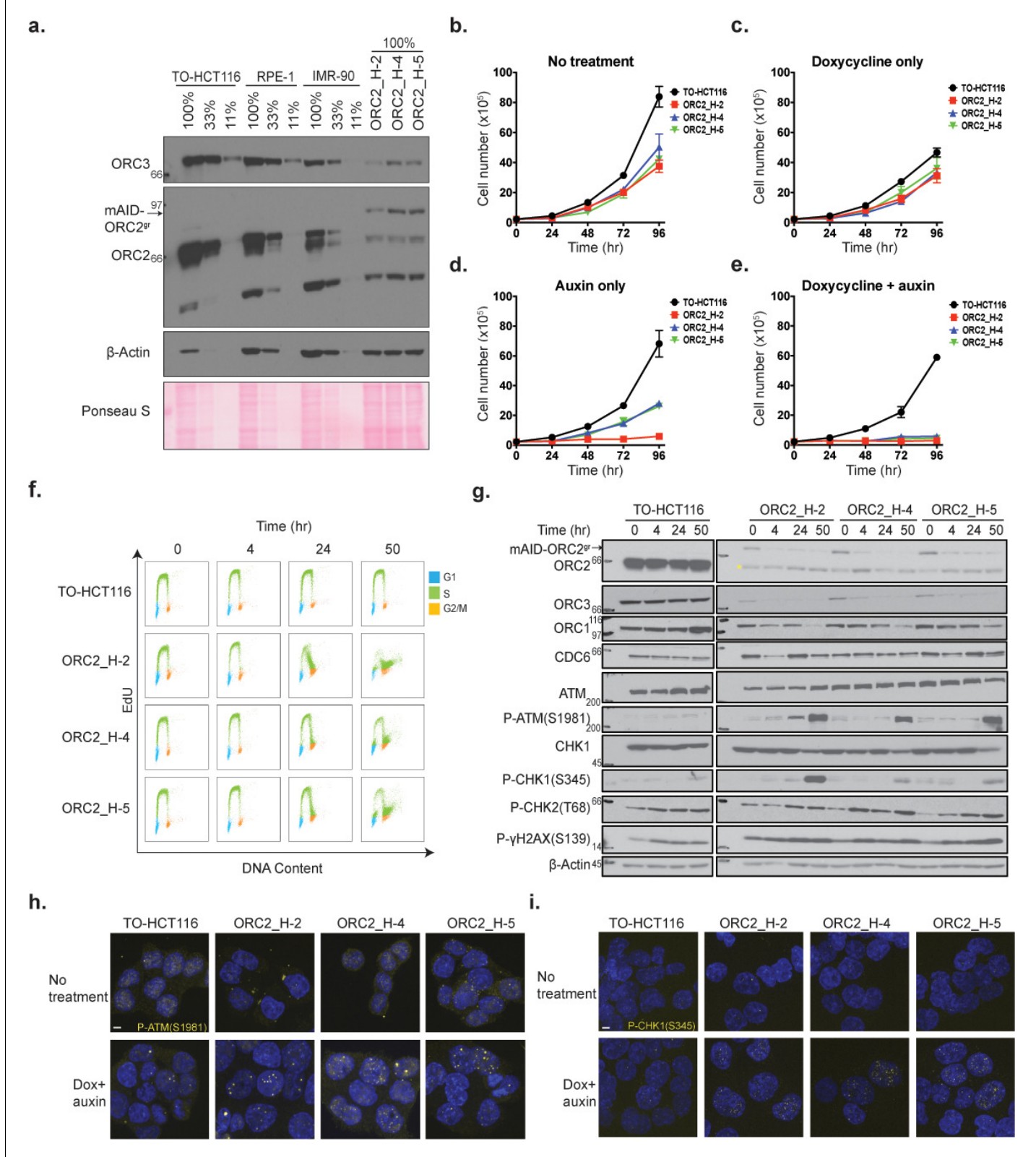

**Figure 3.** Characterization of CRISPR/Cas9 ORC2 knockout and complementation with sgRNA resistant ORC2. (**a**) ORC2, mAID-ORC2$^{gr}$, and ORC3 protein levels in human TO-HCT116, RPE-1, human diploid IMR-90, and three ORC2 KO cell lines. See *Figure 3—source data 1* for uncropped images. (**b–e**) Growth curves of cell lines under (**b**) normal condition, (**c**) doxycycline only, (**d**) auxin only, and (**e**) dox+auxin containing media, respectively. The x axis indicates hours after addition of doxycycline or auxin if any. The y axis reflects the cell number (x10$^5$). n = 3 (biological repeats). Error bars, mean ±

*Figure 3 continued on next page*

*Figure 3 continued*

SD. (f) Cell cycle analysis of TO-HCT116, ORC2_H-2, ORC2_H-4, and ORC2_H-5 cell lines following mAID-ORC2$^{gr}$ depletion. Cells were treated with 0.75 μg/ml of doxycycline for 24 hr before auxin treatment. Cells were pulse labeled with 10 μM EdU for 2 hr before harvesting at 0, 4, 24, and 50 hr time points. The x axis indicates DNA content, and the y axis represents EdU incorporation. Color legend for cell cycle phases - G1-blue; S-green; G2/M-orange. >10,000 cells were analyzed per condition. (g) Protein expression profiles of mAID-ORC2$^{gr}$, ORC2, ORC1, ORC3, CDC6, ATM, p-ATM (S1981), CHK1, p-CHK1(S345), p-CHK2(T68), and p-γH2AX(S139) in four cell lines after dox and auxin treatment for 0, 4, 24, and 50 hr. Cells were treated with doxycycline for 24 hr prior to auxin treatment. Asterisks (*) indicates the non-specific band detected in mutant cell lines. Immunoblot of each protein was developed on the same film at the same time for comparison between all four cell lines. Quantification of ORC1 and CDC6 levels are shown in *Figure 3—figure supplement 3b and c*. See *Figure 3—source data 2* for original uncropped immunoblots. (h) Immunofluorescence staining of p-ATM(S1981) in four cell lines with or without dox+auxin treatment. Quantification of p-ATM(S1981) foci is shown in *Figure 3—figure supplement 3d*. See *Figure 3—source data 3* for uncropped images. (i) Immunofluorescence staining of p-CHK1(S345) in four cell lines with or without dox+auxin treatment. For (h) and (i), dox+auxin-treated cells were stained after incubation with doxycycline for 24 hr followed by addition of auxin for 48 hr. Quantification of p-CHK1(S345) foci is shown in *Figure 3—figure supplement 3e*. See *Figure 3—source data 4* for uncropped images. Scale bar indicated 4 μM.

The online version of this article includes the following source data and figure supplement(s) for figure 3:

**Source data 1.** Entire films of the cropped western blots in *Figure 3a*.
**Source data 2.** Entire films of the cropped western blots in *Figure 3g*.
**Source data 3.** Uncropped immunofluorescence image of *Figure 3h*.
**Source data 4.** Uncropped immunofluorescence image of *Figure 3i*.
**Figure supplement 1.** Validation of CRISPR/Cas9 knockout in ORC2_H-2 cell line.
**Figure supplement 2.** Cell cycle analysis after dox and auxin treatment in TO-HCT116, ORC2_H-2, ORC2_H-4, and ORC2_H-5 cell lines.
**Figure supplement 3.** DNA damage checkpoint is activated in auxin-treated ORC2_H-2, ORC2_H-4, and ORC2_H-5 cell lines.
**Figure supplement 1—source data 1.** Entire films of the cropped western blots in *Figure 3—figure supplement 3a*.
**Figure supplement 1—source data 2.** Uncropped immunofluorescence image of *Figure 3—figure supplement 3f*.

*supplement 1c*). When no auxin was added, ORC1 levels in all four cell lines were similar, but the level gradually decreased after mAID-ORC2$^{gr}$ depletion, which could be an effect of the phase in which cells ultimately arrest at the 50 hr time point (*Figure 3—figure supplement 3b*). On the other hand, CDC6 protein level increased 1.3- to 1.7-fold in mutant cell lines at 0 hr time point, indicating cells might favor higher CDC6 level when ORC is low (*Figure 3—figure supplement 3c*). Phosphorylation of ATM(S1981), ATR(T1989), and CHK1(S345) were detected in H-2, H-4, and H-5 cell lines after 50 hr of auxin treatment, but not in the parental TO-HCT116 (*Figure 3g*, *Figure 3—figure supplement 3a*). Higher levels of P-γH2AX(S139) in H-2, H-4, and H-5 cells were detected even when no auxin was added (*Figure 3g*). This showed that although cells can divide with only 5–10% of ORC2, a certain degree of replication stress exists. In our experiments, when parental and ORC2 mutant cells were treated with doxycycline and auxin, phospho-CHK2(T68) level increased slightly in all four cell lines, suggesting the phosphorylation could be associated with the drug treatment but not depletion of mAID-ORC2$^{gr}$ itself. These observations were also supported by results from immunofluorescent staining of individual cells. When cells were treated with doxycycline and auxin for 48 hr, substantially more ATM(S1981) and CHK1(S345) phosphorylation were detected in all three ORC2 mutant cell lines (*Figure 3h,i*, *Figure 3—figure supplement 3d–e*). In the absence of doxycycline and auxin, the P-γH2AX(S139) signal was more abundant in ORC2_H-2, H-4, and H-5 cells than in wild type (*Figure 3—figure supplement 3f*). To conclude, insufficient ORC2 protein in cells resulted in abnormal DNA replication and DNA damage, and in response to DNA damage, CHK1 was activated and cells arrested in G2 phase.

## Loss of ORC2 results in heterochromatin decompaction and abnormal nuclear morphology

The ORC2 depleted ORC2_H-2 and ORC2_H-5 cells had twice the nuclear volume following treatment with doxycycline and auxin for 48 hr (*Figure 4a–b*) compared to cells without treatment. The average volume of nuclei was greater than the volume of the largest parental cells, and thus the large nuclear phenotype could not be explained by their arrest in G2 phase with a 4C DNA content. However, since ORC2 depletion using siRNA decondenses centromere-associated α-satellite DNA (*Prasanth et al., 2010*), this phenotype could be due to cells arrested in late S and G2 phase with decompacted heterochromatin. During interphase, ORC2 and ORC3 localize to the heterochromatin foci and interact with heterochromatin protein 1α (HP1α) through ORC3 (*Pak et al., 1997*;

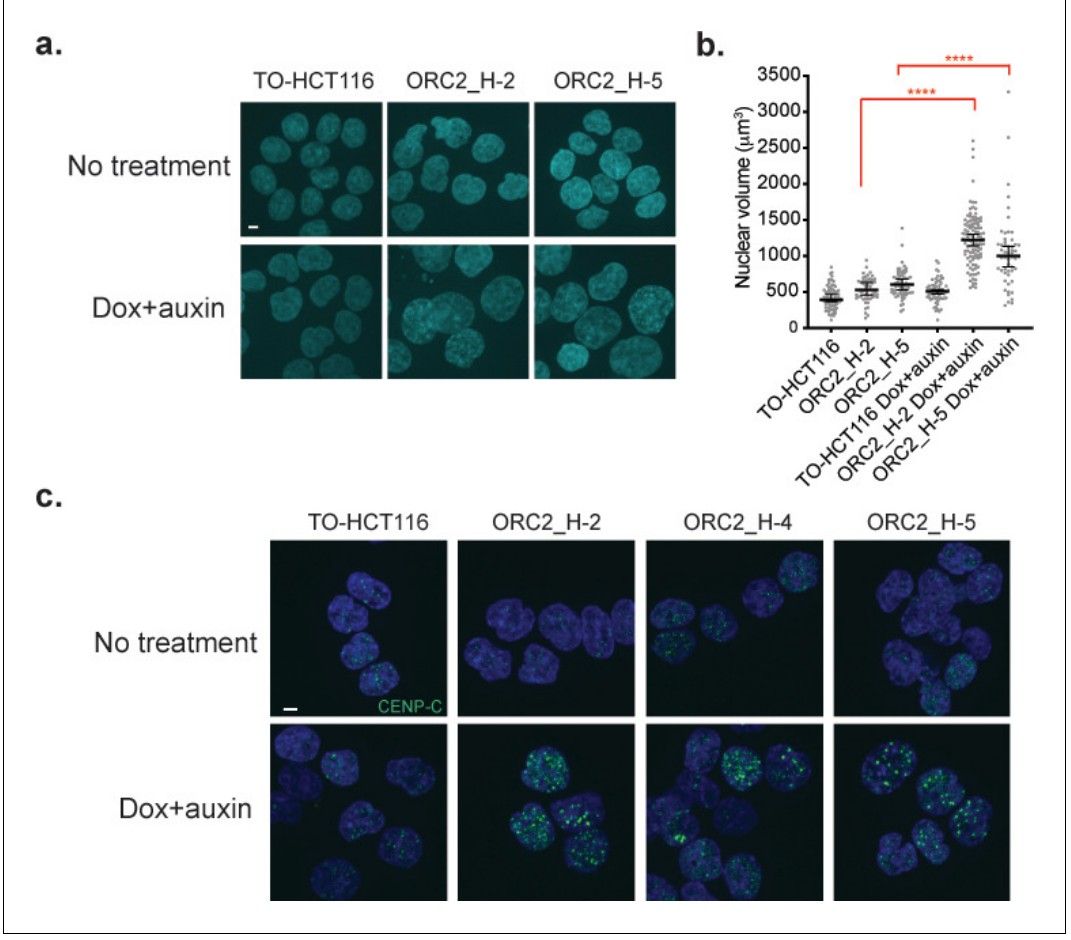

**Figure 4.** Auxin-treated ORC2_H2, H-4, and H-5 cells have abnormal nuclear phenotypes. (a) Nuclear morphology of TO-HCT116, ORC2_H-2, and H-5 cells after 48 hr of auxin treatment. Scale bar indicated 4 µM. See *Figure 4—source data 1* for uncropped images. (b) Scatter plot illustrating the nuclear volume after 48 hr of auxin treatment. Untreated: TO-HCT116, n = 77; ORC2_H-2, n = 52; ORC2_H-5, n = 63. Dox and auxin treated: TO-HCT116, n = 66; ORC2_H-2, n = 110; ORC2_H-5, n = 54. Error bars, medium ± 95% CI. Nuclear volume decreased significantly in both dox and auxin-treated ORC2_H-2 and H-5 cells. Statistical analysis was performed using Student's t-test: ****$p < 0.0001$. All singlets in each field captured have been measured, resulting in different but unbiased sample size selection. See *Figure 4—source data 2* for numerical data table. (c) Immunofluorescence staining of CENP-C after mAID-ORC2$^{gr}$ depletion for 50 hr. Scale bar is 4 µM. See *Figure 4—source data 3* for uncropped images. The online version of this article includes the following source data and figure supplement(s) for figure 4:

**Source data 1.** Uncropped immunofluorescence image of *Figure 4a*.
**Source data 2.** Numerical data table for nuclear volume of *Figure 4b*.
**Source data 3.** Uncropped immunofluorescence image of *Figure 4c*.
**Figure supplement 1.** Quantification of CENP-C foci.
**Figure supplement 2.** Centromeric foci and heterochromatin are decondensed in ORC2-depleted ORC2_H-2 and H-5 cells.
**Figure supplement 3.** Palbociclib synchronization of TO-HCT116 and ORC2_H-2 cell lines.

*Pflumm and Botchan, 2001*; *Prasanth et al., 2004*). To detect heterochromatin decompaction, immunofluorescent staining of the centromeric protein C (CENP-C) and HP1α was performed. In TO-HCT116 cells, CENP-C staining showed multiple, compact foci, but in the doxycycline and auxin treated cells that were dependent on mAID-ORC2$^{gr}$, CENP-C foci were larger and more prominent (*Figure 4c*, *Figure 4—figure supplement 1a–b*). In doxycycline-treated control TO-HCT116,

ORC2_H-2, and ORC2_H-5 cells, normal HP1α foci were observed and most of them localized adjacent to the CENP-C foci. However, in the doxycycline and auxin treated ORC2 mutant cells, HP1α foci were decompacted, shown by an expanded pattern of staining rather than discrete foci found in the parental cells (*Figure 4—figure supplement 2a–f*). This phenotype was observed in cells that also showed the large CENP-C foci phenotype. HP1α is involved in phase separation of heterochromatin (*Larson et al., 2017*; *Strom et al., 2017*), and removing ORC (ORC2 in this case) from the heterochromatin might affect the overall compaction. These HP1α fluorescence patterns were different from those previously observed when ORC was removed using an siRNA approach in HeLa cells (*Prasanth et al., 2010*; *Prasanth et al., 2004*). This difference is most likely because HeLa, being hyper tetraploid cells, have a different nuclear organization of HP1α compared to that found in HCT116 cells. HeLa cell HP1α foci are more intense and discrete and colocalize with CENP-C (*Figure 4—figure supplement 2g*).

To understand if the heterochromatin decompaction phenotype was a direct outcome of the loss of ORC2, we synchronized cells using Palbociclib, a CDK4/6 inhibitor that arrests cells in G1 phase. TO-HCT116 and ORC2_H-2 cells were incubated with Palbociclib and doxycycline for 28 hr, with auxin added during the last 4.5 hr in G1 phase, then cells were harvested after a further 12 hr incubation (*Figure 4—figure supplement 3a*). The second harvest time point was 40 hr after Palbociclib treatment and 16.5 hr after auxin treatment. The nuclear volume was determined, and we found that it remained unchanged in both cell types (*Figure 4—figure supplement 3b*). CENP-C and HP1α staining was also unaffected (data not shown). Thus, loss of ORC2 in G1 phase did not induce nuclear chromatin decompaction. One possibility is that the absence of ORC2 during S phase affects sister chromatid cohesion loading (*Zheng et al., 2018*), which leads to decompaction of centromeric and heterochromatin regions in late S/G2 phase.

## ORC2 is essential for initiation of DNA replication

When cells were treated with siRNA against *ORC2* for 72 hr, 30% of the cells arrested in a mitosis-like state (*Prasanth et al., 2004*). This observation led to the conclusion that ORC2 is not only required for the initiation of DNA replication, but also during mitosis. To examine the role of ORC2 in G1 and mitosis following acute depletion, TO-HCT116 and the ORC2_H-2 cells were synchronized at the G1-early S phase boundary with a 2C DNA content by a double thymidine block, with doxycycline being added during the second thymidine block. Where indicated, auxin was added 4.5 hr before the release from the second thymidine block (*Figure 5a*). Synchronized cells were then released and allowed to progress through two cell division cycles. Cells were pulse labeled with EdU for 2 hr and harvested at several timepoints after release. The mAID-ORC2$^{gr}$ protein was depleted upon release from second thymidine block, and the depletion persisted until the final collection at 48 hr in the auxin-treated ORC2_H-2 cells (*Figure 5—figure supplement 1a*, lane 23–28).

During the first cell cycle following release into S phase, no obvious change in DNA content and EdU labeling was observed in both cell lines, irrespective of whether or not they were treated with doxycycline and auxin (*Figure 5b–c*; *Figure 5—figure supplement 1b*). During the second cell cycle, however, doxycycline and auxin-treated TO-HCT116 cells progressed through S phase only slightly slower than the untreated cells. By contrast, starting from the second cell cycle, serious defects were observed in ORC2_H-2 auxin-treated cells compared to untreated cells (*Figure 5b–c*). First, auxin-treated ORC2_H-2 cells exhibited a very slow S phase, indicating that cells were struggling to completely replicate their DNA. Second, cells arrested with a 4C DNA content, which could either be late S or G2/M phase. Third, after 48 hr following release from the double thymidine block, some cells arrested in G1 phase and could not enter S phase. We suggest that in the first cell cycle, although mAID-ORC2$^{gr}$ was depleted just before release, the pre-RCs were already formed and the cells were primed to replicate the complete genome. However, starting from the second cell cycle, since ORC2 was entirely lost from cells following cell cycle progression through the previous S, G2 and M phase, the cells began to accrue cell cycle defects. Most ORC2-depleted cells then either arrested in G1 or went through an incomplete S phase and arrested at the G2/M phase, but did not progress further. This experiment indicated ORC2 primarily functions in establishing DNA replication initiation, but based on the results thus far, we could not conclude a role during mitosis because in the first cycle ORC2 depleted cells with a 4C DNA content, progressed through G2/M phase and cell division into the next G1.

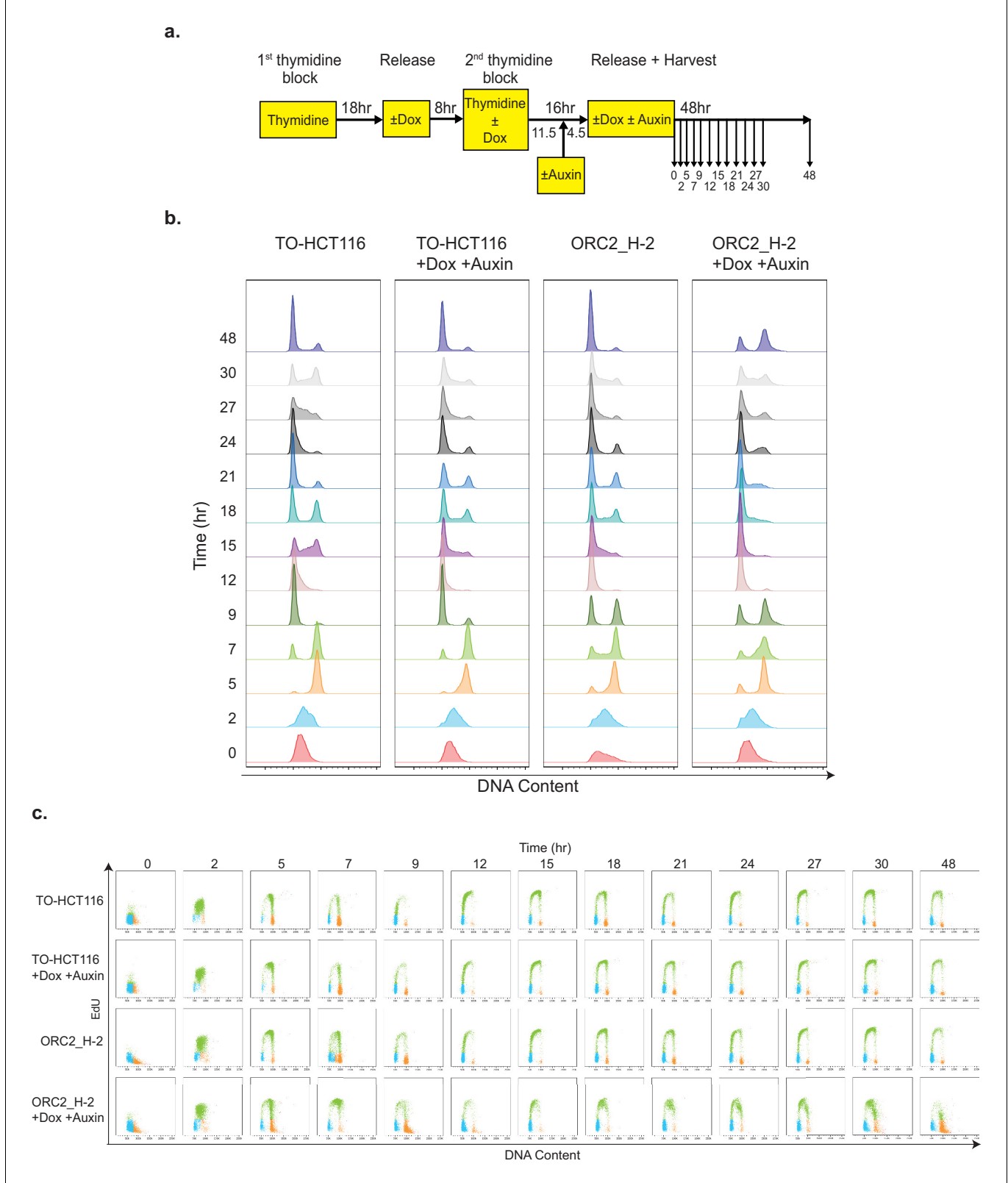

**Figure 5.** ORC2_H-2 cells show abnormal cell cycle progression after mAID-ORC2[gr] depletion. (a) Experimental scheme of TO-HCT116 and ORC2_H-2 cells synchronization by a double thymidine block. (b) Flow cytometry analysis of FxCycle Violet stained cells (singlets) released from double thymidine block in indicated treatment. (c) Cell cycle profiles of TO-HCT116 and ORC2_H-2 cells released from a double thymidine block in indicated treatment. *Figure 5 continued on next page*

Cells were pulse labeled with 10 µM EdU for 2 hr before harvesting at different time points. X axis indicates DNA content, and y axis represents to EdU incorporation. Color legend for overlay plots of cell cycle phases - G1-blue; S-green; G2/M-orange.

The online version of this article includes the following figure supplement(s) for figure 5:

**Figure supplement 1.** Double thymidine block and release in TO-HCT116 and ORC2_H-2 cell lines.

## An MCM complex loading and pre-RC assembly defect in ORC2 depleted cells

The auxin-treated, mAID-ORC2$^{gr}$-depleted cells could not replicate normally, possibly due to insufficient ORC to form the pre-RC. To test this hypothesis, the chromatin-bound MCM2-7 was measured in asynchronous cells by flow cytometry following extraction by detergents as described previously (*Matson et al., 2017*). Asynchronous TO-HCT116, ORC2_H-2, and H-5 cells, with or without doxycycline and auxin treatment, were pulse-labeled with EdU, harvested and stained with anti-MCM2 antibody and DNA dye. In normal medium and without detergent extraction, nearly 100% of the cells were positive for MCM2 in all three cell lines (*Figure 6—figure supplement 1b*). When extracted with detergent, about 78% of TO-HCT116, 65.2% of ORC2_H-2, and 76.9% of ORC2_H-5 cells were positive for MCM2. Chromatin-bound MCM2 level was not affected in untreated or doxycycline-treated TO-HCT116 and ORC2 mutant cells (*Figure 6a*). When treated with both doxycycline and

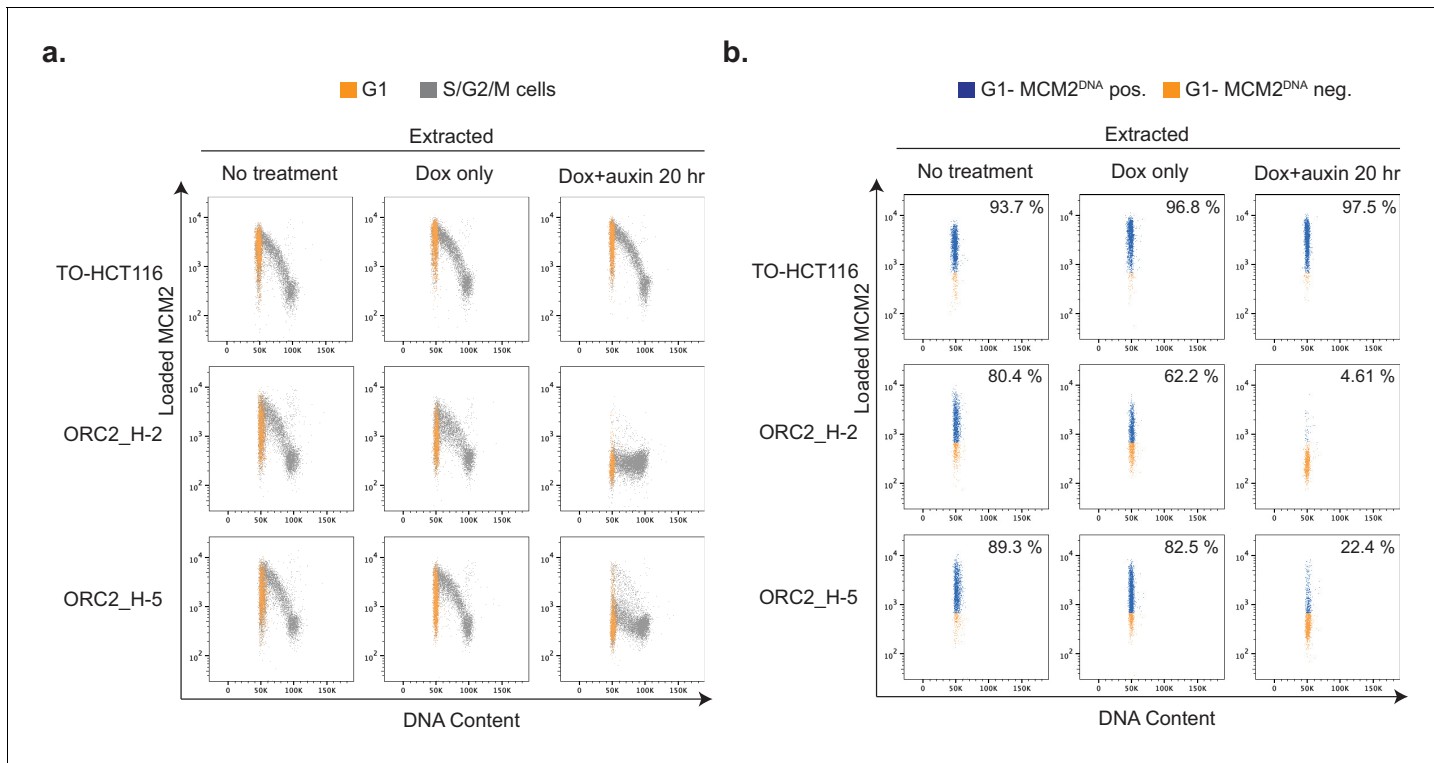

**Figure 6.** Depletion of mAID-ORC2$^{gr}$ in ORC2_H-2 and H-5 cells results in decreased DNA-loaded MCM. (a) Flow cytometry analysis of DNA content and chromatin-bound MCM2 in asynchronous TO-HCT116, ORC2_H-2, and ORC2_H-5 cells in different conditions. The x axis indicates DNA content, and the y axis represents cells positive for chromatin-bound MCM2 as a function of its fluorescence intensity. G1 population (gated from DNA content vs EdU plot (*Figure 6—figure supplement 1c*)) is shown in orange. The rest of the cells in S/G2/M phase are shown in gray. (b) Only G1 cell populations from (a) are shown here, with DNA-bound MCM2 positive cells colored in blue and negative cells in orange. Numbers at the upper right corner indicates percentage of MCM2 positive cells. The x axis indicates DNA content, and the y axis represents cells positive for chromatin-bound MCM2 as a function of its fluorescence intensity.

The online version of this article includes the following figure supplement(s) for figure 6:

**Figure supplement 1.** MCM2 binding to chromatin detected using flow cytometry.

**Figure supplement 2.** Chromatin-loaded MCM2 decrease after mAID-ORC2$^{gr}$ depletion.

auxin for 20 hr, chromatin-bound MCM2 levels decreased significantly in mutant cells, especially in the ORC2-H-2 cell line (*Figure 6a*, *Figure 6—figure supplement 1a,c*). Focusing on the G1 population, 93.7% of TO-HCT116 cells, 80.4% of ORC2_H-2 cells, and 89.3% of ORC2_H-5 cells were positive for MCM2 (*Figure 6b*) in untreated cells. When doxycycline was added, MCM2 level in ORC2 H-2 cells dropped slightly to 62.2%, while the other two cell lines remained at similar level, 96.8% in TO-HCT116% and 82.5% in ORC2_H-5 cells. When treated with both doxycycline and auxin, 97.5% of TO-HCT116 G1 phase cells stained for chromatin-bound MCM2, but this decreased to only 4.6% and 22.4% cells in ORC2_H-2 and ORC2_H-5 G1 phase cells, respectively. Importantly, the doxycycline and auxin treated ORC2_H-2 and ORC2 H-5 G1 phase cells had very low levels of MCM2 compared to the MCM2 levels in cells that did not receive auxin (*Figure 6b*). The different degrees of reduction in MCM2 loading between doxycycline and auxin treated H-2 and H-5 cells reflected the relative levels of mAID-ORC2$^{gr}$. Therefore, ORC2 depletion prevented MCM2 loading onto G1 phase chromatin.

We also analyzed chromatin-bound MCM2 levels in TO-HCT116, ORC2_H-2, and ORC2_H-5 cells following a double thymidine block and release experiment (*Figure 6—figure supplement 2a*). Flow cytometry analysis showed that at the 0 hr time point with cells arrested at the G1-early S boundary, the percentage of cells with chromatin-bound MCM2 were unchanged, whether or not doxycycline or auxin was added. At 12 hr time point, with the cell cycle mostly back at G1 phase again, cells with chromatin-bound MCM2 level decreased significantly in doxycycline and auxin-treated ORC2_H-2 and ORC2_H-5 cells (*Figure 6—figure supplement 2*). This result also confirmed that since pre-RCs were already formed during the double thymidine block, ORC2 depletion did not have an effect, but new pre-RCs could not form in the next cell cycle.

## ORC2 depletion in cells leads to aberrant mitosis

In order to know if the mAID-ORC2$^{gr}$-depleted ORC2_H-2 cells entered mitosis, we evaluated the mitotic index by staining cells for phospho-Histone H3(S10) (pH3S10) followed by flow cytometry analysis (*Figure 7a*; *Figure 7—figure supplement 1*, n = 3 biological repeats). In the untreated asynchronous population, about 4.53 ± 0.59% and 1.57 ± 0.33% were pH3S10-positive in TO-HCT116 and ORC2_H-2 cells respectively, while 31.4 ± 2.88% of TO-HCT116 cells and 15.6 ± 1.25% of ORC2_H-2 cells were at G2/M. When only doxycycline was added, there was no significant change. When treated with doxycycline and auxin for 28 hr, the pH3S10-positive cell population percentage was about 2.39 ± 0.26 in TO-HCT116 and only 0.79 ± 0.09 in ORC2_H-2, while 17.23 ± 0.78% of TO-HCT116 cells and 36.7 ± 1.61% of ORC2_H-2 cells were at G2/M. After 50 hr of doxycycline and auxin treatment, the pH3S10-positive cell population percentage was 3.95 ± 0.16 in TO-HCT116 and only 0.96 ± 0.15 in ORC2_H-2, while 20.77 ± 1.76% of TO-HCT116 cells and 79.57 ± 1.2% of ORC2_H-2 cells were at G2/M phase. In normal medium condition, TO-HCT116 already had 2.9 times as many mitotic cells as ORC2_H-2. When treated with doxycycline and auxin, although the G2/M population increased 2.3-fold and fivefold at 28 hr and 50 hr, respectively, the number of mitotic cells in ORC2_H-2 reduced by 50–80% compared to the non-treated H-2 cells. This showed that most ORC2_H-2 cells that accumulated at the 4C DNA peak after ORC2 depletion were indeed stuck in the G2 stage and did not enter into mitosis.

Nevertheless, mitosis did occur at a very low frequency. To observe mitotic phenotypes and mitotic progression following ORC2 depletion, we constitutively expressed H2B-mCherry in TO-HCT116 and ORC2_H-2 cells via lentiviral transduction and performed time lapse fluorescent imaging of the mitotic chromosomes. Cells were synchronized using a single thymidine block and auxin was added or omitted 2 hr before releasing into fresh medium with or without doxycycline and auxin. As expected, in either treated or untreated TO-HCT116 cells, the first cell cycle after release from the thymidine block was normal and cells progressed through mitosis into the second cell cycle (data not shown). During the second cell cycle, in the absence or presence of doxycycline and auxin, it took TO-HCT116 cells about 35–50 min to progress from early prophase to chromosome segregation (*Figure 7b and c*, n = 10 for each). In the absence of doxycycline and auxin, ORC2_H-2 cells also progressed through mitosis like the parental cells (*Figure 7d*, n = 10). In stark contrast, in presence of doxycycline and auxin ORC2_H-2 cells showed condensed chromatin and attempted to congress chromosomes at the metaphase plate but never achieved correct metaphase alignment of chromosomes even after 6–13 hr (*Figure 7e*, n = 5). Those few cells that did attempt anaphase had

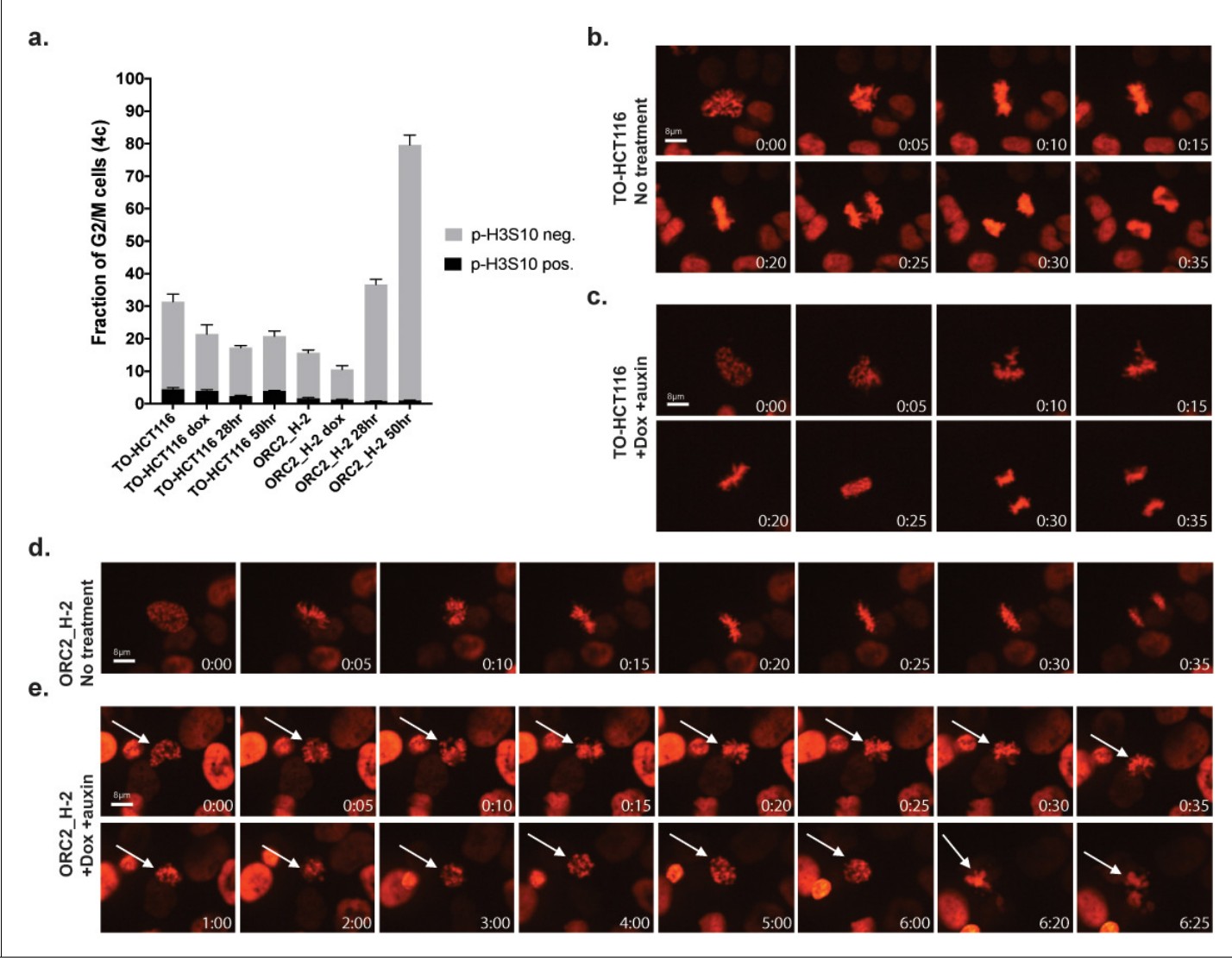

**Figure 7.** ORC2_H-2 cells have aberrant mitosis after auxin treatment. (**a**) Mitotic index of TO-HCT116 and ORC2_H-2 G2/M cells with or without auxin. 0.75 µg/ml Doxycycline were added for 24 hr before auxin treatment. Cells were harvested after 0, 28, or 50 hr of auxin treatment followed by staining with anti-pH3S10 antibody for mitotic cells and FxCycle Violet for DNA content. Histograms on x axis represent each cell line under different conditions, including no treatment, doxycycline only, dox+auxin for 28 hr, and dox+auxin for 50 hr. The y axis is the fraction of 4C G2/M cells. Cell population positive or negative for p-H3S10 were shown as black or gray color respectively. n = 3 (biological repeats). See *Figure 7—source data 1* for numerical data. (**b–e**) Time lapse imaging of TO-HCT116 and ORC2_H-2 cell lines following a single thymidine block (± dox) and release into the second cell cycle (i.e. >30 hr following release). Time shown in lower left corner indicates time (hr: min) since early prophase. (**b**) Images of TO-HCT116 cells without auxin. See *Figure 7—source data 2* for uncropped images. (**c**) Auxin treated TO-HCT116 cells. See *Figure 7—source data 3* for uncropped images. (**d**) ORC2_H-2 cells without auxin. See *Figure 7—source data 4* for uncropped images. (**e**) Dox and auxin treated ORC2_H-2 cells. White arrows in (**e**) point to the same cell. Scale bars indicate 8 µM. See *Figure 7—source data 5* for uncropped images.

The online version of this article includes the following source data and figure supplement(s) for figure 7:

**Source data 1.** Numerical data table for p-H3S10 flow cytometry in *Figure 7a*.
**Source data 2.** Uncropped immunofluorescence image of *Figure 7b*.
**Source data 3.** Uncropped immunofluorescence image of *Figure 7c*.
**Source data 4.** Uncropped immunofluorescence image of *Figure 7d*.
**Source data 5.** Uncropped immunofluorescence image of *Figure 7e*.
**Figure supplement 1.** Flow cytometry gating strategy for *Figure 7a*.

abnormal chromosome segregation, producing lagging chromosomes, micronuclei and became apoptotic.

## Characterization of previously published *ORC1$^{-/-}$* and *ORC2$^{-/-}$* cell lines

The results so far confirm previous observations that ORC is essential for cell proliferation in differentiated human cells, but there remained the curious case of previously reported viable knockout of *ORC1* and *ORC2* genes in p53$^{-/-}$ HCT116 cells (*Shibata et al., 2016*). We obtained the *ORC1$^{-/-}$* (B14 clone) and *ORC2$^{-/-}$* (P44 clone) cell lines described in this study as a gift from Dr. Anindya Dutta and performed several experiments on them. Using the ORC2 validated sgRNAs (different target site from Shibata et al.), we first tested if the *ORC2$^{-/-}$* cell line was sensitivity to the four *ORC2* sgRNAs we used in our study (*Figure 2e*). The negative-selection GFP depletion assay surprisingly showed that both the parental HCT116 *p53$^{-/-}$* and the *ORC2$^{-/-}$* cell line were sensitive to CRISPR/Cas9 knockout of *ORC2* compared to control sgRNA (*Figure 8a–b*). More importantly, when both cell lines were transduced to express mAID-ORC2$^{gr}$ that was resistant to the ORC2-1 and ORC2-2 sgRNAs, this effect was rescued in HCT116 *p53$^{-/-}$* and to a slightly lesser extent in *ORC2$^{-/-}$* cell line (*Figure 8c–d*). This suggested that there may be some form of functional ORC2 in the *ORC2$^{-/-}$* cells that was being targeted by the tested sgRNAs. In addition, an immunoblot of the cell lysates showed a reduced level of ORC3 in the *ORC2$^{-/-}$* cells (*Figure 8—figure supplement 1a*), and since ORC2 and ORC3 form stable heterodimers in cells, this result again indicated that some form of ORC2 was expressed in cells, albeit at a lower level. When immunoprecipitated with an antibody against ORC3, we detected ORC3 and a putative truncated form of ORC2 which was seen only in *ORC2$^{-/-}$* cells (*Figure 8—figure supplement 1b*). Next, we designed primer pairs that span exon junctions for each exon in *ORC2* and performed quantitative RT-PCR (qPCR) to determine the nature of the *ORC2* transcripts in the *ORC2$^{-/-}$* cells (*Figure 8e*). The calculated fold change (FC) indicated that in the *ORC2$^{-/-}$* cells, about 60% of the mRNAs had exon seven skipped, whereas other exons remained the same (*Figure 8e*, *Figure 8—figure supplement 2*).

We further speculated that in these cell lines, the absence of or mutations within either *ORC1* or *ORC2* may exhibit genomic instability giving rise to copy number variations (CNVs). To determine the CNV status we performed SMASH (*Wang et al., 2016*) analysis on the two parental HCT116 cell lines with the *p53$^{+/+}$* and *p53$^{-/-}$* background as well as the Shibata et al. *ORC1$^{-/-}$* and *ORC2$^{-/-}$* deficient lines. Both the parental cell lines showed very similar chromosome copy number, characteristic of HCT116, while both *ORC1* deficient and *ORC2*-deficient cell lines had additional CNVs in chromosomes unrelated to those harboring either *ORC1* or *ORC2* (*Figure 8—figure supplement 3*). The significance of these specific loci which showed alterations in copy number when either *ORC1* or *ORC2* was deleted remains to be investigated. However, it was in this analysis that we noticed that in *ORC2$^{-/-}$* cells a part of the *ORC2* gene locus that was hugely amplified (*Figure 8—figure supplement 3* solid arrow). To study in detail the *ORC2* gene region on chromosome two in these cells, we performed long-read Oxford Nanopore sequencing analysis. Compared to the reference allele sequence, *ORC2$^{-/-}$* cells showed highly mutated and heterogenous allele distribution near the CRISPR targeting site used in the Shibata et al. study (*Figure 8f*, *Figure 8—figure supplement 4*). In addition, this *ORC2$^{-/-}$* cell line contained many copies of the plasmid with ORC2 homology arms, used for integration of a disruptive blasticidin cassette into the ORC2 gene locus (*Shibata et al., 2016*). In fact, it was the *ORC2* homology arms in the plasmid that showed up as the massive amplification in the CNV analysis by SMASH (*Figure 8—figure supplement 4a,b,c*). Based on RT-PCR data, aside from the expected heterozygous deletions in exon 7, exon 4 also exhibited multiple heterozygous SNPs at three different sites. Among them, SNP1 resulted in a novel stop codon, while SNP3/4/5 were missense mutations, and SNP2/6 were silent mutations (*Figure 8f*). Next, we determined the haplotype phasing information between the heterozygous exon 7 deletion and the potential novel stop codons. One allele of ORC2 contained various mutations in exon 4, in combination with the deleted exon 7, showing that the cell line was heterogeneous to start with. The other allele of *ORC2* that had an intact exon seven also had either wild type or SNPs in exon 4. Since there is a methionine in exon 5, it is possible the truncated ORC2 protein we observed was translated from this internal start site (*Figure 8—figure supplement 1b*). Although substantially altered, based on the sequencing data and the sensitivity of the cells to ORC2-1 and ORC2-2 sgRNAs, we conclude that the cell strain is not a true *ORC2* knockout.

With regard to the HCT116 $ORC1^{-/-}$ cell line, we confirmed that they lacked ORC1 protein using multiple antibodies and confirmed that they duplicated at a much slower rate than the parental line, as previously reported (*Figure 8—figure supplement 1c*; *Shibata et al., 2016*). The doubling time of the $ORC1^{-/-}$ cells was four-times longer than the parental cells and we were unable to passage these cells for many generations (by 20–30 generations they stopped proliferating). We also compared HCT116 $ORC1^{-/-}$ (B14) and HCT116 $ORC2^{-/-}$ (P44) cell lines with the parental lines, that is either $p53^{+/+}$ and $p53^{-/-}$ background by confocal microscopy [*Figure 8g* (1-4), 8 hr; *Figure 8—figure supplement 5a–e*, *Figure 8—figure supplement 6a–c*]. There were a myriad of nuclear morphology defects in the $ORC1^{-/-}$ cell line. When the nuclei were stained with Hoechst dye, up to 10% contained abnormally large nuclei or sometimes what seemed to be multiple nuclei aggregated together in single cell, while other cells appeared normal. When probed further by staining for F-actin and Lamin B1 to observe overall cellular morphology and nuclear membrane integrity respectively, we observed that despite the staining for DNA content looking normal, up to 50% of the $ORC1^{-/-}$ cells showed highly abnormal, involuted nuclear membranes (*Figure 8i*). In addition, most of the gigantic nuclei seemed to have lost the nuclear membrane altogether, while those cells that had Lamin B1 staining displayed abnormal nuclear membrane integrity.

The chromatin organization in the cells was observed by transmission electron microscopy (TEM) and revealed huge differences in cell size and nuclear structure between the wild type HCT116 $p53^{+/+}$ and $ORC1^{-/-}$ cells (*Figure 8—figure supplement 6d–f*). About 35% of $ORC1^{-/-}$ cells were grossly larger than wild type cells. Those multinucleate/polyploid giant cells were full of membrane invagination and vacuoles, and also exhibit significant apoptotic activities. Most likely they were formed due to extensive DNA damage and nuclear structural defects and underwent a different type of cell division called neosis, in which intracellular cytokinesis occurs and some mononuclear cells are produced from nuclear budding or asymmetric cell division (*Sundaram et al., 2004*). All these phenotypes pointed to the fact that although $ORC1^{-/-}$ cells do not survive in culture long term, for the duration that they do grow, they proliferate extremely slowly and are grossly abnormal. It may well be the case that $p53^{-/-}$ status of the parental HCT116 was required for these cells to be produced in the first place.

## Discussion

The $ORC2^{-/-}$ cell line believed to be a complete knockout via the use of 3 sgRNAs, one targeting the exon 4, and the others targeting the sixth and seventh introns retained a truncated form of ORC2 that could interact with ORC3 and was expressed from a mutated gene. These cells were still susceptible to $ORC2$ knockdown using four sgRNAs selected from our CRISPR screens and also partially rescued the phenotype with two sgRNAs using a CRISPR-sgRNA resistant mAID-ORC2$^{gr}$. Similar to what we found for $ORC2^{-/-}$ cells, CRISPR-induced frameshifts in cells often generate truncated proteins that, although may not be recognized by western blot, still preserve whole or partial protein function (*Smits et al., 2019*). Based on these observations with ORC2 and the results with $ORC1^{-/-}$ lines, cells were unable to proliferate for many generations and produced abnormally structured cells, as well as data analyzed by tiling-sgRNA CRISPR screens, we conclude that ORC is essential in human cells. This conclusion is consistent with existing literature (*Hemerly et al., 2009*; *McKinley and Cheeseman, 2017*; *Ohta et al., 2003*; *Prasanth et al., 2010*; *Prasanth et al., 2004*; *Prasanth et al., 2002*) and is not surprising since ORC has multiple functions in human cells.

The fact that $ORC1^{-/-}$ cells could be obtained might suggest that ORC1 is not essential, but these cells did proliferate very slowly and only for a limited number of generations, and yielded cells with grossly abnormal nuclear structures. Thus, we suggest that it is unlikely to obtain cells that are viable over many generations in the absence of all ORC subunits. The $ORC1^{-/-}$ cells have excess levels of CDC6 protein bound to chromatin (*Shibata et al., 2016*) and since CDC6 is related to ORC1 and can bind to the other ORC subunits, we suggest that CDC6 might compensate, albeit poorly, for the absence of ORC1 in loading MCM2-7 proteins and establishing pre-RCs. A recent report described human cells that proliferated in the absence of ORC2 or ORC5 (*Shibata and Dutta, 2020*), but we suggest that a detailed molecular and phenotypic analysis of these cells be examined in greater detail as described in this report.

The pooled CRISPR/Cas9 domain-focused screen has become a common and powerful tool for uncovering genes that are essential for cell proliferation, cell survival, and for identification of

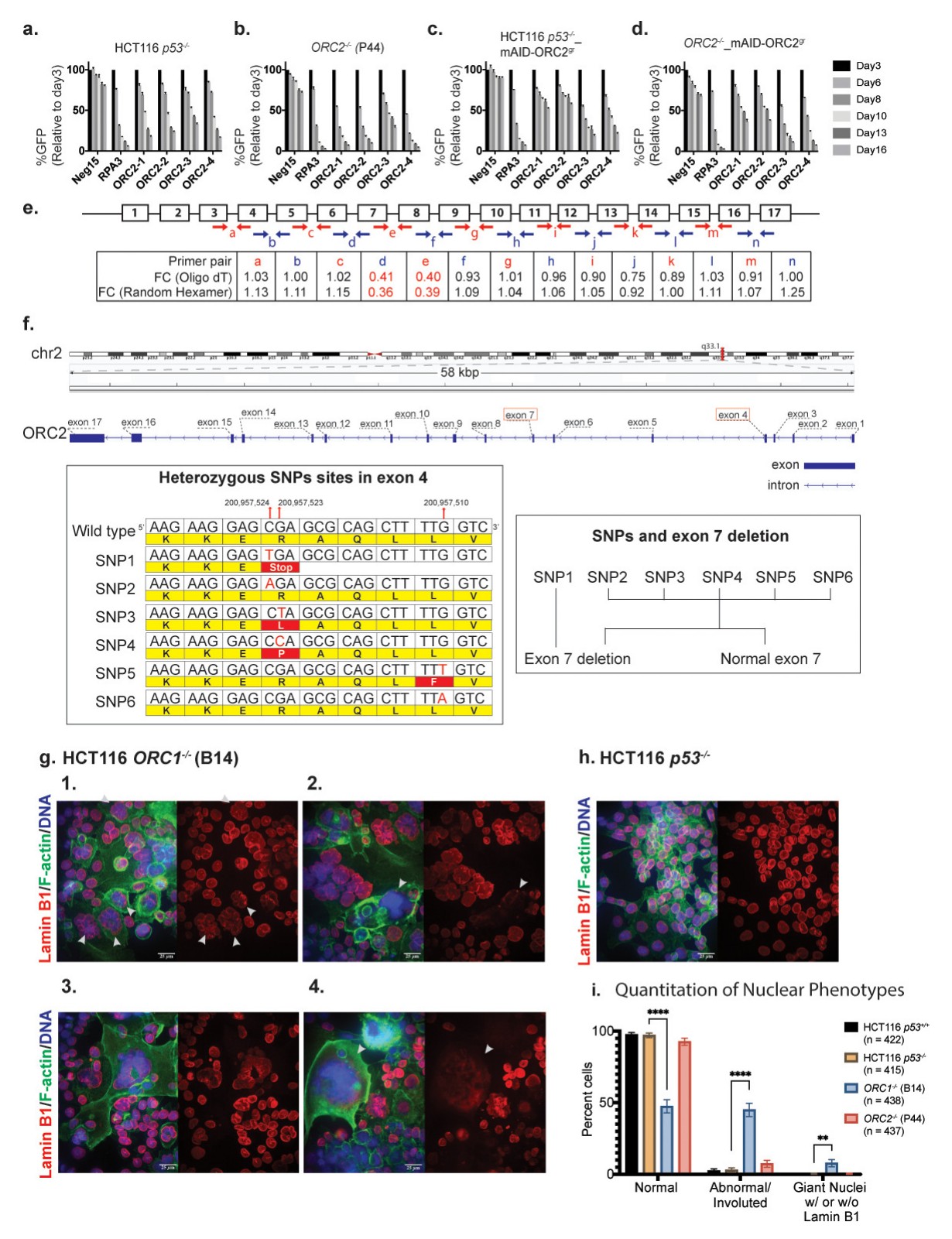

**Figure 8.** Characterization of previously published *ORC1⁻/⁻* and *ORC2⁻/⁻* cell lines. (a–d) Negative-selection time course assays that plot the percentages of GFP positive cells over time following transduction with the indicated sgRNAs/Cas9. Experiments were performed in HCT116 *p53⁻/⁻*, *ORC2⁻/⁻*, HCT116 *p53⁻/⁻*_mAID-ORC2^gr, and *ORC2⁻/⁻*_mAID-ORC2^gr cell lines. The GFP positive percentage was normalized to the Day3 measurement. n = 3. Error bars, mean ± SD. (e) Calculated fold change (FC) for each primer pairs in *ORC2⁻/⁻* cells compared to HCT116 *p53⁻/⁻* cells. The red and blue arrows

*Figure 8 continued on next page*

*Figure 8 continued*

indicate each primer pair. Two kinds of primers, Oligo dT and Random Hexamer, were used in the reverse transcription step. Bar diagram view is shown in *Figure 8—figure supplement 1*. (f) Structural variations (SVs) in the *ORC2* gene. Three SNP sites were among the six heterozygous mutations found in the fourth exon. Heterozygous deletion of exon 7 is also found in *ORC2$^{-/-}$* cells. Long ONT reads that span both the heterozygous deletion site in exon 7 and the heterozygous SNP site in exon 4 show that SNP1 is on the same haplotype that contains the deletion of exon 7. The other haplotype contains a complete copy of exon seven with heterozygous SNPs in exon 4. (g) 1–4: Immunofluorescence of HCT116 *ORC1$^{-/-}$* (B14) cell line stained with Anti-Lamin B1 antibody (Red), Phalloidin (F-actin) (Green), Hoechst Dye (Blue). Images show either merge of all three channels or Lamin-B1 staining of the nuclei. White arrows indicate abnormal and involuted nuclei in image g1. White arrows also show extremely large (nuclear giants) that have lost nuclear membrane integrity (g2, g4). (h) Parental cell line for the *ORC1$^{-/-}$* line as representative control for quantitative and qualitative comparison. More fields of control cells HCT116 *p53$^{+/+}$* and *p53$^{-/-}$* background and *ORC1$^{-/-}$* and *ORC2$^{-/-}$* cell lines are shown in *Figure 8—figure supplements 5* and *6*. Scale bar is 25 µm (i) Quantitation of abnormal nuclei between cell lines. Nuclei per field were classified as Normal, Abnormal/Involuted or Nuclear giants (with or without Lamin B1). Multiple fields were counted to classify > 400 cells for each cell line (n = sample size indicated in legend). Significance calculated using two-way ANOVA for multiple comparisons keeping HCT116 *p53$^{-/-}$* as control. ****p<0.0001, **p=0.0014 (See *Figure 8— source data 1*).

The online version of this article includes the following source data and figure supplement(s) for figure 8:

**Source data 1.** Numerical data table and statistical analysis for graph in *Figure 8i*.
**Figure supplement 1.** ORC3 exists in *ORC2$^{-/-}$* cell line.
**Figure supplement 2.** Real time quantitative PCR fold change represented as bar plot (for *Figure 8e*).
**Figure supplement 3.** Copy number analysis of the genomes of four cell lines using the SMASH method.
**Figure supplement 4.** Amplified regions of the *ORC2* gene and associated transfected DNA in the *ORC2$^{-/-}$* P44 clone.
**Figure supplement 5.** Confocal microscopy images of HCT116 cell lines.
**Figure supplement 6.** Confocal (a–c) and Transmission electron microscopy (d–f) (TEM) images of HCT116 cell lines.

essential functional domains in proteins (*Adelmann et al., 2018*; *Park et al., 2017*; *Shi et al., 2015*; *So et al., 2019*). However, if the screens use only a handful of guides targeting annotated essential regions, it may still result in data which may or may not score a gene as essential. Tiling-sgRNA CRISPR-Cas9 screens on the other hand test 'functional' or 'essential' domains in a more rigorous and unbiased way (*He et al., 2019*). Using this approach, sgRNAs tiled across entire open reading frames of ORC1-6 and CDC6 enabled us to correlate the negative selection phenotype to functional domains within these proteins. The combined results also confirmed that all ORC1-6 and CDC6 proteins were essential in cancer cells as well as a human diploid cell line, including ORC2 that was characterized as non-essential based on multiple shRNA and whole genome CRISPR/Cas9 screens in numerous types of cells in the DepMap portal (https://depmap.org/portal/). We were able to identify many sgRNAs that targeted *ORC2* in the tiling-sgRNA CRISPR screen and the two chosen cloned sgRNAs that killed cells were successfully complemented using a *mAID-ORC2$^{gr}$* transgene, demonstrating specificity of the knockdowns. Thus, single-guide experiments, especially those with negative results, should be interpreted with caution, such that the essential nature of a gene should be examined in depth as we have done here.

The known functional domains in ORC1, including the BAH, AAA+, and WHD were identified using the open reading frame tiling-sgRNA CRISPR/Cas9 screen, as well as other regions of ORC1, including the intrinsically disordered region (IDR; amino acids 180–480, *Figure 1e*) which we know binds to Cyclin A-CDK2 and CDC6 (*Hossain et al., 2019*), as well as many other proteins we have identified and characterized in detail. Additionally, this entire IDR may contribute to DNA-mediated ORC liquid phase transition (*Hossain et al., 2019*; *Parker et al., 2019*). The screen also identified an essential region of ORC1 in and around amino acid 750–790 (*Figure 1a–b*), which may represent the pericentrin-AKAP450 centrosomal targeting (PACT) domain that localizes ORC1 to centrosomes to regulate correctly centrosome and centriole copy number (*Hemerly et al., 2009*).

In ORC2, multiple, essential domains were identified, including the AAA+-like domain and the WHD. The WHD of human ORC2 controls access of human ORC to DNA by inserting itself into the DNA-binding channel prior to activation of the protein by binding to ORC1 and subsequent binding to CDC6 (*Bleichert, 2019*; *Hossain et al., 2019*; *Jaremko et al., 2020*). The ORC2-carboxy terminus binds to ORC3 and ORC2 is also known to bind to PLK1, the mitotic protein kinase (*Song et al., 2011*). Interestingly, ORC2 also has an IDR (*Figure 2—figure supplement 1d*; amino acids 30–230)

and the sgRNA tiling screen of this region shows CKHS essential amino acids, but a relatively conserved region within this IDR amino acid is reproducibly essential in both HCT116 and RPE-1 cell lines (*Figure 2a–b* and *Figure 2—figure supplements 1a* and *9a–e*). Recent studies have implicated the N-terminal region of ORC2 to interact with ORC associated (ORCA) protein and may functionally contribute to its role in DNA replication and chromatin organization (*Shen et al., 2012*).

The use of a mAID-ORC2$^{gr}$ enabled rapid removal of ORC2 from cells and analysis of the resulting phenotypes in more detail. It was not surprising that ORC2 is essential for loading MCM2, and hence MCM2-7, to establish pre-RCs and origins of DNA replication across the genome. In the absence of ORC2, cells loaded very little MCM2, most likely resulting in too few origins of replication and a consequently slow S phase and arrest with a near 4C DNA content and ongoing DNA synthesis. ORC2 depletion yielded other phenotypes, including large nuclei and a failure to execute mitosis. The large nuclei, also observed in the *ORC1*$^{-/-}$ cells, have large CENP-C and HP1$\alpha$ foci, probably due to decompaction of the centromeric associated $\alpha$-satellite DNA, as observed previously (*Prasanth et al., 2010*). We suggest a general role for ORC in nuclear organization and organizing chromatin domains in the nucleus, including heterochromatin. In yeast, ORC is essential for transcriptional silencing at the silent mating type heterochromatic loci *HMRa* and *HML$\alpha$* loci and its function in replication are separable from that in silencing (*Bell et al., 1993*; *Palacios DeBeer et al., 2003*; *Ehrenhofer-Murray et al., 1995*). In *Drosophila*, ORC localizes and associates with heterochromatin protein HP1 during interphase and mitosis, and heterozygous recessive lethal mutations in *DmORC2* suppress position effect variegation (*Huang et al., 1998*; *Pak et al., 1997*). In humans, ORC1 interacts with RB and SUV39H1, a histone methyltransferase that tri-methylates histone H3K9 which HP1 binds to repress E2F1-dependent CCNE1 transcription (*Hossain and Stillman, 2016*). ORC1 and ORC3 (a tight ORC2 binding partner) directly interact with HP1, and depletion of ORC subunits disrupt localization of HP1 and the compaction of chromosome 9 $\alpha$-satellite repeats DNA (*Pak et al., 1997*; *Prasanth et al., 2010*). Furthermore, ORC1 binds to the histone H3K9me3 and H4K20me2 methylation marks and co-localizes with histone H2A.Z, suggesting ORC may organize higher order chromatin structures via direct interactions with modified histones (*Hossain and Stillman, 2016*; *Kuo et al., 2012*; *Long et al., 2019*). The mechanism by which the nuclei become large as a result of ORC depletion is under further investigation.

A final phenotype we observed in the acute removal of ORC2 is that the cells that replicate DNA and enter into mitosis attempt chromosome congression at the metaphase plate, but never make it, even after 7 hr and eventually die of apoptosis. We had observed abnormal mitotic cells following long-term (72 hr) treatment of cells with shRNAs that targeted *ORC2* but it was not clear if this phenotype was due to incomplete DNA replication (*Prasanth et al., 2004*). However, in the current study, acute removal of ORC2 captured some cells with a clear defect in chromosome congression during mitosis. Moreover, both ORC2 and ORC3 localize to centromeres (*Craig et al., 2003*; *Prasanth et al., 2004*), suggesting that they play a role in spindle attachment or centromeric DNA organization, particularly the centromere associated satellite repeat sequences. We speculate that in ancestral species, ORC localized at origins of DNA replication and this ORC also functioned in organization of chromosomes and in chromosome segregation, but upon separation of DNA replication and chromosome segregation with the advent of mitosis, separate functions of ORC in DNA replication, chromatin, or nuclear organization and chromosome segregation were retained, but executed at different times during the cell division cycle.

## Materials and methods

**Key resources table**

| Reagent type (species) or resource | Designation | Source or reference | Identifiers | Additional information |
|---|---|---|---|---|
| Gene (*Homo sapiens*) | ORC1 | GenBank | NM_004153.4 | |
| Gene (*Homo sapiens*) | ORC2 | GenBank | NM_006190.5 | |
| Gene (*Homo sapiens*) | ORC3 | GenBank | NM_181837.3 | |
| Gene (*Homo sapiens*) | ORC4 | GenBank | NM_001190879.3 | |

*Continued on next page*

*Continued*

| Reagent type (species) or resource | Designation | Source or reference | Identifiers | Additional information |
|---|---|---|---|---|
| Gene (*Homo sapiens*) | ORC5 | GenBank | NM_002553.4 | |
| Gene (*Homo sapiens*) | ORC6 | GenBank | NM_014321.4 | |
| Gene (*Homo sapiens*) | CDC6 | GenBank | NM_001254.4 | |
| Strain, strain background (*Escherichia coli*) | Stbl3 | NEB | C3040 | High efficiency chemically competent cells |
| Cell line (*H. sapiens*) | HCT116 *p53+/+* | ATCC | Cat# CCL-247, RRID:CVCL_0291 | Cell line maintained in B. Stillman Lab |
| Cell line (*H. sapiens*) | RPE-1 | ATCC | Cat# CRL-4000, RRID:CVCL_4388 | Cell line maintained in B. Stillman Lab |
| Cell line (*H. sapiens*) | HEK293T | ATCC | Cat# CRL-3216, RRID:CVCL_0063 | Cell line maintained in B. Stillman Lab |
| Cell line (*H. sapiens*) | HCT116 *p53-/-* | *Bunz et al., 1998* | RRID:CVCL_S744 | Generous gift from Anindya Dutta (University of Virginia) |
| Cell line (*H. sapiens*) | *HCT116 p53-/- ORC1-/-* (clone B14) | *Shibata et al., 2016* | N/A | Generous gift from Anindya Dutta (University of Virginia) |
| Cell line (*H. sapiens*) | *HCT116 p53-/- ORC2-/-* (clone P44) | *Shibata et al., 2016* | N/A | Generous gift from Anindya Dutta (University of Virginia) |
| Cell line (*H. sapiens*) | U2OS | ATCC | Cat# HTB-96, RRID:CVCL_0042 | Cell line maintained in B. Stillman Lab |
| Cell line (*H. sapiens*) | TO-HCT116 (Tet-OsTIR1 HCT116) | *Natsume et al., 2016* | N/A | Generous gift from Masato T. Kanemaki (National Institute of Genetics, Japan) |
| Cell line (*H. sapiens*) | ORC2_H-2 | This study | N/A | Cell line derived from TO-HCT116 |
| Cell line (*H. sapiens*) | ORC2_H-4 | This study | N/A | Cell line derived from TO-HCT116 |
| Cell line (*H. sapiens*) | ORC2_H-5 | This study | N/A | Cell line derived from TO-HCT116 |
| Antibody | Lamin B1; Rabbit polyclonal | Abcam | Cat# ab16048, RRID:AB_10107828 | IF – 0.2 µg/ml |
| Antibody | ORC1, mouse monoclonal (pKS1-40) | CSHL In-house | N/A | IB – 1:1000 |
| Antibody | ORC2, rabbit polyclonal (CS205) | CSHL In-house | N/A | IB – 1:10,000 |
| Antibody | ORC3, rabbit polyclonal (CS1980) | CSHL In-house | N/A | IB – 1:10,000 |
| Antibody | CDC6, mouse monoclonal (DCS-180) | EMD Millipore | Cat# 05–550 RRID:AB_2276118 | IB – 1:1000 |
| Antibody | ATM, rabbit monoclonal (Y170) | Abcam | Cat# ab32420 RRID:AB_725574 | IB – 1:1000 |
| Antibody | p-ATM(S1981), rabbit monoclonal (EP1890Y) | Abcam | Cat# ab81292 RRID:AB_1640207 | IB – 1:1000 IF – 1:200 |
| Antibody | CHK1, rabbit monoclonal (EP691Y) | Abcam | Cat# ab40866 RRID:AB_726820 | IB – 1:1,000 |
| Antibody | p-CHK1(S345), rabbit monoclonal (133D3) | Cell Signaling | Cat# 2348 RRID:AB_331212 | IB – 1:1000 IF – 1:200 |
| Antibody | p-CHK2(T68), rabbit monoclonal (C13C1) | Cell Signaling | Cat# 2197 RRID:AB_2080501 | IB – 1:1000 |

*Continued on next page*

*Continued*

| Reagent type (species) or resource | Designation | Source or reference | Identifiers | Additional information |
|---|---|---|---|---|
| Antibody | p-γH2AX(S139), rabbit monoclonal (20E3) | Cell Signaling | Cat# 9718 RRID:AB_2118009 | IB – 1:1000 IF – 1:200 |
| Antibody | ATR, rabbit polyclonal | Abcam | Cat# ab2905 RRID:AB_303400 | IB – 1:1000 |
| Antibody | p-ATR(T1989), rabbit polyclonal | Abcam | Cat# ab227851 (discontinued) | IB – 1:1000 |
| Antibody | p-ATR(S428), rabbit polyclonal | Cell Signaling | Cat# 2853 RRID:AB_2290281 | IB – 1:1000 |
| Antibody | β-Actin, mouse monoclonal (8H10D10) | Cell Signaling | Cat# 3700 RRID:AB_2242334 | IB – 1:10,000 |
| Antibody | CENP-C, Mouse monoclonal (2159C5a) | Abcam | Cat# ab50974 RRID:AB_869095 | IF – 1:200 |
| Antibody | HP1α, Mouse monoclonal (2HP-1H5) | Millipore | Cat# MAB3584 RRID:AB_94938 | IF – 1:500 |
| Antibody | ECL anti-Rabbit IgG Horseradish Peroxidase linked whole antibody | GE Healthcare | Cat# NA934V | IB – 1:10,000 |
| Antibody | ECL anti-mouse IgG Horseradish Peroxidase linked whole antibody | GE Healthcare | Cat# NA931V | IB – 1:10,000 |
| Antibody | Goat Anti-Mouse IgG H and L Alexa Fluor 647 | Abcam | Cat# ab150115 RRID:AB_2687948 | IF – 1:1000 |
| Antibody | Goat Anti-Rabbit IgG H and L Alexa Fluor 488 | Abcam | Cat# ab150077 RRID:AB_2630356 | IF – 1:1000 |
| Antibody | Goat Anti-Rabbit IgG H and L (Alexa Fluor 594) | Abcam | Cat# ab150084, RRID:AB_2734147 | IF – 1:1000 |
| Antibody | MCM2 (BM28); mouse monoclonal | BD Biosciences | Cat #610700 RRID:AB_2141952 | FC – 1:200 |
| Antibody | Alexa Fluor 647 donkey anti-mouse antibody | Jackson Immuno Research Labs | Cat# 715-605-151, RRID:AB_2340863 | FC – 1:1000 |
| Antibody | Phospho-Histone H3 (Ser10), Mouse monoclonal (6G3) | Cell Signaling Technology | Cat# 9706, RRID:AB_331748 | FC – 1:25 |
| Recombinant DNA reagent | LentiV_Cas9_puro (plasmid) | Addgene | RRID:Addgene_108100 | Lentiviral expression of cDNA with puromycin resistance gene – used for making RPE-1 cas9 puro (generous gift from Jason Sheltzer, CSHL) |
| Recombinant DNA reagent | LentiV_Cas9_Blast (plasmid) | Addgene | RRID:Addgene_125592 | Lentiviral expression of cDNA with blasticidin resistance gene – used for making HCT116 cas9 blast (generous gift from Chris Vakoc CSHL) |
| Recombinant DNA reagent | LRG2.1 (plasmid) | Addgene | RRID:Addgene_108098 | BsmBI digestion for sgRNA cloning |
| Recombinant DNA reagent | LgCG_cc88 lentiviral vector (plasmid) | N/A | N/A | Lentiviral expression of Cas9-sgRNA-GFP – used for dropout CRISPR/Cas9 experiment (generous gift from Chris Vakoc, CSHL) |
| Recombinant DNA reagent | epCas9-1.1-mCherry (plasmid) | *Chang et al., 2020* | N/A | Generous gift from David Spector (CSHL) |

*Continued on next page*

*Continued*

| Reagent type (species) or resource | Designation | Source or reference | Identifiers | Additional information |
|---|---|---|---|---|
| Recombinant DNA reagent | pHAGE-CMV-MCS-IZsGreen (plasmid) | N/A | N/A | Lentiviral expression vector – used to construct pHAGE-CMV-H2B-mCherry |
| Recombinant DNA reagent | mAID-mCherry2-NeoR (plasmid) | Addgene | RRID:Addgene_72830 | The plasmid was used to construct mAID-ORC2 transgene |
| Recombinant DNA reagent | pMSCV-hygro retroviral (plasmid) | TaKaRa | Cat #634401 | Retroviral expression of cDNA with hygromycin resistance gene – used to construct pMSCV-hygro-mAID-ORC2 to express mAID-ORC2 in cells |
| Transfected construct (*H. sapiens*) | pMSCV-hygro-mAID-ORC2 (plasmid) | This study | N/A | Retroviral construct for transduction and express mAID-ORC2 |
| Transfected construct (*H. sapiens*) | sgRNA_ORC2-1-epCas9-1.1-mCherry (plasmid) | This study | N/A | Construct to transfect and express Cas9 and sgRNA ORC2-1 in human cells |
| Transfected construct (*H. sapiens*) | pHAGE-CMV-H2B-mCherry (plasmid) | Gift from Dr. Alea Mills, Cold Spring Harbor Laboratory | N/A | Lentiviral construct for transduction and express H2B-mCherry in human cells |
| Sequence-based reagent | F2 | This paper | PCR primer for amplification of sgRNA cassette | TCTTGTGGAAAGG ACGAAACACCG |
| Sequence-based reagent | R2 | This paper | PCR primer for amplification of sgRNA cassette | TCTACTATTCTTTCC CCTGCACTGT |
| Commercial assay or kit | NEBuilder HiFi DNA Assembly Cloning Kit | NEB | Cat# E5520S | |
| Commercial assay or kit | Click-iT EdU Alexa Fluor 488 Flow Cytometry Assay Kit | Invitrogen | Cat# C10420 | |
| Commercial assay or kit | DNeasy Blood and Tissue kit | Qiagen | Cat# 69504 | |
| Commercial assay or kit | RNeasy Mini kit | Qiagen | Cat# 74104 | |
| Software, algorithm | Volocity 3D Image Analysis Software | Perkin Elmer | RRID:SCR_002668 | |
| Software, algorithm | GraphPad Prism 9 | GraphPad | RRID:SCR_002798 | |
| Software, algorithm | Model-based Analysis of Genome-wide CRISPR-Cas9 Knockout (MAGeCK) | Li, et al. MAGeCK enables robust identification of essential genes from genome-scale CRISPR/Cas9 knockout screens. Genome Biology 15:554 (2014) | N/A | https://sourceforge.net/p/mageck/wiki/Home/ |
| Software, algorithm | Protiler Analysis | He et al. De novo identification of essential protein domains from CRISPR-Cas9 tiling-sgRNA knockout screens. Nat Commun 10, 4547 (2019) | N/A | https://github.com/MDhewei/protiler |

*Continued on next page*

*Continued*

| Reagent type (species) or resource | Designation | Source or reference | Identifiers | Additional information |
|---|---|---|---|---|
| Software, algorithm | FlowJo | BD | RRID:SCR_008520 | |
| Software, algorithm | ImageJ | NIH | RRID:SCR_003070 | |
| Chemical compound, drug | Palbociclib | Selleckchem | Cat# S1116 | 1 μM |
| Chemical compound, drug | Thymidine | Millipore Sigma | Cat# 89270 | 2 mM |
| Chemical compound, drug | Doxycycline | CalBiochem | Cat# 324385 | 0.75 ug/ml |
| Chemical compound, drug | Auxin (Indole-3-acetic acid sodium salt) | Millipore Sigma | Cat# 15148 | 500 nM |
| Chemical compound, drug | DAPI | Life Technologies | Cat# D1306 | 1 μg/ml |
| Chemical compound, drug | FxCycle Violet Stain | ThermoFisher | Cat# F10347 | 1:1000 |
| Chemical compound, drug | Hoechst dye | ThermoFisher | Cat# 62249 | 1 μg/ml |
| Chemical compound, drug | Phalloidin iFluor 488 | Abcam | Cat# ab176753 | 1:1000 |
| Chemical compound, drug | Polyethylenimine (PEI 25000) | Polysciences | Cat# 23966–100 | 1 mg/mL |

## Cell culture

HCT116 (WT *p53*$^{+/+}$), U2OS, RPE-1, Plat-E cells and HEK293T cell lines were cultured in DMEM (Gibco) supplemented with 10% Fetal bovine serum and 1% Penicillin/Streptomycin. IMR-90 cell line was cultured in EMEM supplemented with 10% Fetal bovine serum and 1% Penicillin/Streptomycin. Plat-E and HEK293T cells were used for retroviral and lentiviral production, respectively. HCT116 (*p53*$^{-/-}$), HCT116 *ORC1*$^{-/-}$ (*p53*$^{-/-}$ background, clone B14), HCT116 *ORC2*$^{-/-}$ (*p53*$^{-/-}$ background, clone P44) were a kind gift from Dr. Anindya Dutta (University of Virginia, Charlottesville, VA, USA). Tet-OsTIR1 HCT116 (TO-HCT116) cell line was a kind gift from Dr. Masato Kanemaki (National Institute of Genetics, Mishima, Japan). All gifted cell lines were cultured in McCoys 5A (Gibco) supplemented with 10% fetal bovine serum and 1% Penicillin/Streptomycin. All cell lines were cultured at 37°C with 5% $CO_2$. All the cell lines used in this study were tested for mycoplasma and were negative.

## Tiling-sgRNA guide design

Every possible guide directly upstream of a sp-Cas9 canonical PAM (NGG) sequence in the 5'- > 3' direction was extracted from the target exon sequences. Guides with the canonical PAM (NGG) were aligned to the GRCh38 genome using the BatMis exact k-mismatch aligner (*Tennakoon et al., 2012*). A maximum of three mismatches were considered for off-target evaluation. The resulting alignment file was parsed, and each off-target location assigned a penalty according to the number of mismatches to the target sequence and the exact position of each mismatch in the guide, where the farther the mismatch is from the PAM the higher the penalty, and based on the proximity of the mismatches to each other; assigning higher penalties to mismatches that are farther apart.

The resulting penalties from each assessed off-target site were then combined into a single off-target score for each guide (*Hsu et al., 2013*) with 1.00 as the maximum possible score for guides not having any off-target site with up to three mismatches. The final results included the guide sequence, the PAM, the number of off-target sites in the genome with 0, 1, 2, and 3 mismatches, the cut site location, the calculated off-target score, and any RefSeq genes (*O'Leary et al., 2016*) located at the off-target sites. All guides, including positive and negative controls used in this study are reported in *Supplementary file 1*_guides.

## Plasmid construction and sgRNA cloning

HCT116-Cas9 and RPE-1-Cas9 expressing cell lines were a gifts from Dr. Chris Vakoc and Dr. Jason Sheltzer, respectively (Cold Spring Harbor Laboratory, NY, USA). In this study, all the sgRNAs targeting genes of interest as well as controls were cloned into LRG2.1 plasmid (derived from U6-sgRNA-GFP, Addgene: 108098) - as described (*Tarumoto et al., 2018*). Single sgRNAs were cloned by annealing sense and anti-sense DNA oligos followed by T4 DNA ligation into a BsmB1-digested LRG2.1 vector. To improve U6 promoter transcription efficiency, an additional 5' G nucleotide was added to all sgRNA oligo designs that did not already start with a 5' G.

For an unbiased tiling-sgRNA CRISPR screen, pooled sgRNA libraries were constructed. All designed sgRNAs including positive/negative controls were synthesized in duplicate or triplicate in a pooled format on an array platform (Twist Bioscience) and then PCR cloned into the Bsmb1-digested LRG2.1 vector using Gibson Assembly. To ensure the representation and identity of sgRNA in the pooled lentiviral libraries, a MiSeq analysis was performed (Illumina) and we verified that 100% of the designed sgRNAs were cloned in the LRG2.1 vector and that the abundance of > 95% of the sgRNA constructs was within fivefold of the mean.

For *ORC2* CRISPR complementation assays, sgRNA resistant synonymous mutations were introduced to ORC2 by PCR mutagenesis using Phusion high fidelity DNA polymerase (NEB). Amplified Guide RNA-resistant ORC2 (ORC2gr) was cloned into NheI-digested mAID-mCherry2-NeoR plasmid (mAID-mCherry2-NeoR, Addgene 72830) to add mAID degron sequence to the N-terminus. The mAID-ORC2gr was then PCR amplified and assembled into BglII/XhoI digested pMSCV-hygro retroviral vector (TaKaRa #634401). Cloning was done using In-Fusion cloning system (TaKaRa). In this experiment, sgRNAs targeting ORC2 and control sgRNAs were cloned into BsmB1digested LgCG_cc88 lentiviral vector (Vakoc laboratory, CSHL) by the same sgRNA cloning strategy described above.

To create *ORC2* CRISPR/Cas9 knock out TO-HCT116 cells, we used sgRNA_ORC2-1-epCas9-1.1-mCherry plasmid for transient transfection. Sequence of sgRNA_ORC2-1 was cloned into epCas9-1.1-mCherry plasmid which was a kind gift from Dr. David Spector (Cold Spring Harbor Laboratory, NY, USA). sgRNAs were cloned by annealing sense and anti-sense DNA oligos followed by T4 DNA ligation into a BbsI-digested epCas9-1.1-mCherry vector.

To construct a lentiviral vector that constitutively expresses H2B-mCherry in TO-HCT116 and ORC2_H-2 cells, H2B-mCherry sequence was PCR amplified and cloned into BamHI/BspDI -digested pHAGE-CMV-MCS-IZsGreen vector which was a kind gift from Dr. Alea Mills (Cold Spring Harbor Laboratory, NY, USA).

## Viral transductions

Lentiviruses were produced in HEK293T cells by co-transfecting target plasmid and helper packaging plasmids psPAX2 and pVSVG with polyethylenimine (PEI 25000, Polysciences cat# 23966–100) transfection reagent. Plasmids were mixed in the ratio of 1:1.5:2 of psPAX2, pVSVG and target plasmid DNA in OptiMEM (Gibco, Cat# 31985062). 1 mg/mL PEI was added, mixed, and incubated, before addition to the cells. Cell culture medium was changed 7 hr after transfection, and viral supernatant collected at 36 and 72 hr following transfection. For the high-throughput lentiviral screening, viral supernatant was concentrated with Lenti-X Concentrator (Takara, #631231) following manufacturer's protocol.

Retroviruses were produced in Plat-E cells by co-transfecting target plasmid and packaging plasmids pCL-Eco and pVSVG in the ratio of 1.25:1:9 with PEI. Cell culture medium was changed 7 hr after transfection, and the supernatant was collected at 36 hr post-transfection.

For either lenti or retroviral transductions, target cells were mixed with viral supernatant, supplemented with 8 µg/mL polybrene and centrifuged at 1700 rpm for 30 min at room temperature. Fresh medium was added 24 hr after transduction. Where selection was required, antibiotics – 1 µg/mL puromycin; 10 µg/mL of blasticidin; 200 µg/ml of hygromycin – were added 72 hr post infection.

## Pooled sgRNA screening

CRISPR-based negative selection screens using sgRNA libraries targeting proteins ORC1-6, CDC6 as well as positive and negative controls, were performed in stable Cas9-expressing HCT116 ($p53^{+/+}$) and RPE-1 cell lines. The screens were performed as previously described (*Lu et al., 2018*;

*Miles et al., 2016*; *Shi et al., 2015*) with a few optimizations for scale. Briefly, to ensure a single copy sgRNA transduction per cell, multiplicity of infection (MOI) was set to 0.3–0.35. To achieve the desired representation of each sgRNAs during the screen, the total number of cells infected was determined such that while maintaining the MOI at ~ 0.3, each guide would yield at least 2000 counts at the beginning with Illumina NGS. Cells were harvested at day three post-infection and served as the initial time-point (P1) of the pooled sgRNA library, representing all guides transduced to begin with. Cells were cultured for 10 population doublings (P10) and harvested as the final time point. Genomic DNA was extracted using QIAamp DNA midi kit (QIAGEN) according to the manufacturer's protocol. Data from a total of 3 screens (HCT116: n = 2; RPE-1: n = 1) is presented in this study.

Next-generation sequencing (NGS) library was constructed based on a newly developed protocol. To quantify the sgRNA abundance at P1 and P10, the sgRNA cassette was PCR amplified from genomic DNA using Amplitaq Gold DNA Polymerase (Invitrogen, 4311820) and primers (F2: TCTTG TGGAAAGGACGAAACACCG; R2: TCTACTATTCTTTCCCCTGCACTGT). The resulting DNA fragment (~242 bp) was gel purified. In a second PCR reaction Illumina-compatible P7 and custom stacked barcodes (*Supplementary file 2*) which included the standard Illumina P5 forward primer, were introduced into samples by PCR amplification and the final product was gel purified (~180–200 bp). Samples were quantified by Agilent Bioanalyzer DNA High-sensitivity Assay (Agilent 5067–4626) and pooled together in equal molar ratios and analyzed by Illumina. Libraries were sequenced with a single-end 76 cycle NextSeq 500/550 kit on the NextSeq mid-output platform.

## Quantification and analysis of screen data

The quantification of guides was done using a strict exact match to the forward primer, sample barcode, and guide sequence. MAGeCK was used for the identification of essential sgRNAs by running the 'mageck test' command on the P1 and P10 raw sgRNA counts. MAGeCK employs median normalization followed by a Negative Binomial modeling of the counts, and provides the log fold change (LFC) and p-values at both the individual guide and gene levels (*Li et al., 2014*).

We used Protiler (https://github.com/MDhewei/ProTiler-1.0.0), a computational method for the detection of CRISPR Knockout Hypersensitive (CKHS) regions from high-throughput tiling screens, to call and visualize the essential domains (*He et al., 2019*). Protiler uses denoising methods to mitigate the off-target effects and inactive sgRNAs, then applies a wavelet-based changing point detection algorithm to delineate the boundaries of sensitive regions. We separately input averaged LFC values from the two replicates of HCT116 or RPE-1 computed from MAGeCK, and analyzed the dataset at default values for all parameters except -t2/–threshold2. This threshold detects changing points using TGUH method described in this pipeline and we identified CKHS regions at thresholds of 0.25 and 0.5 for each target protein.

## GFP competition and sgRNA complementation assay

TO-HCT116, TO-HCT116_mAID-ORC2$^{gr}$, U2OS, U2OS_mAID-ORC2$^{gr}$, HCT116 *p53$^{-/-}$*, HCT116 *p53$^{-/-}$*_mAID-ORC2$^{gr}$, *ORC2$^{-/-}$* p44, and *ORC2* p44$^{-/-}$_ mAID-ORC2$^{gr}$ cells were transduced with individual sgRNA-Cas9-GFP lentiviruses at an MOI of 0.3–0.4 to ensure one copy of sgRNA per cell. Cells were passaged every 3 days beginning day 3 (P1) till day 21(P7) post-transduction. At each passage, GFP percentage were evaluated by Guava easyCyte flow cytometer. Three technical repeats were measured for each datapoint. Measured values were normalized to GFP percentages at P1 for each sgRNA. All experiments were performed as a set of 3 biological replicates.

## Generating endogenous ORC2 KO mAID-ORC2$^{gr}$ cell lines

To construct TO-HCT116-mAID-ORC2$^{gr}$ cells, TO-HCT116 cells were transduced with mAID-ORC2$^{gr}$ retrovirus and selected with 200 µg/ml of hygromycin. sgRNA_ORC2-1-epCas9-1.1-mCherry plasmid was transiently transfected into TO-HCT116-mAID-ORC2$^{gr}$ cells using Lipofectamine 2000 Transfection Reagent (ThermoFisher #11668019) following manufacturer's protocol. Cells were harvest by 0.25% trypsin-EDTA after 24 hr, washed once with PBS, and resuspended into sorting buffer containing 2% FBS, 2 mM EDTA, and 25 mM HEPES pH7.0. Single cells were FACS sorted and expanded.

## Cell proliferation assays

TO-HCT116, ORC2_H-2, ORC2_H-4, and ORC2_H-5 cell lines were seeded 24 hr before the experiment in either base medium or medium containing 0.75 μg/ml doxycycline. For each cell line, 150,000 cells were seeded on day 1, and medium was changed every day. We harvested three replicates for each time point. Cells stained with 0.4% trypan blue solution were counted using an automated cell counter. Similarly, cell proliferation assays for HCT116 $p53^{+/+}$, HCT116 $p53^{-/-}$, $ORC1^{-/-}$ and $ORC2^{-/-}$ cells were done starting at a seeding density of 100,000 cells.

## Immunoprecipitation, Immunoblotting, and quantitation

Cells were incubated in RIPA buffer (150 mM NaCl, 1 % NP-40, 0.5% Sodium deoxycholate, 0.1% SDS, 25 mM Tris-HCl PH 7.4) on ice for 15 min. Laemmli buffer was then added and samples analyzed by western blotting to detect proteins with antibodies. Primary antibodies: anti-ORC2 (rabbit polyclonal #CS205, in-house), anti-ORC3 (rabbit polyclonal #CS1980, in-house), anti-ORC1 (mouse monoclonal #pKS1-40, in-house), anti-CDC6 (mouse monoclonal #DCS-180, EMD Millipore), anti-ATM (rabbit monoclonal #ab32420, abcam), anti-pATM(S1981) (rabbit monoclonal #ab81292, abcam), anti-CHK1 (rabbit monoclonal #ab40866, abcam), anti-pCHK1(S345) (rabbit monoclonal #2348, Cell Signaling), anti-pCHK2(T68) (rabbit monoclonal #2197, Cell Signaling), anti-ATR (rabbit polyclonal #ab2905, abcam), anti-pATR(T1989) (rabbit polyclonal #ab227851, abcam), anti-pATR (S428) (rabbit polyclonal #2853, Cell Signaling), anti-p-γH2AX(S139) (rabbit monoclonal #9718, Cell Signaling), anti-β-Actin (mouse monoclonal #3700, Cell Signaling). Secondary antibodies: ECL anti-Rabbit IgG Horseradish Peroxidase linked whole antibody (#NA934V, GE Healthcare) and ECL anti-mouse IgG Horseradish Peroxidase linked whole antibody (#NA931V, GE Healthcare).

Relative ORC2 (or mAID-ORC2$^{gr}$), ORC3, ORC1, and CDC6 protein levels in each cell line was quantified by normalizing band area to β-Actin of each cell line and then normalized to HCT116 cells using ImageJ software.

## Cell cycle analysis and pulse EdU label

For double-thymidine block and release experiments, cells were first incubated with 2 mM thymidine for 18 hr. After PBS washes, cells were released into fresh medium, with or without (0.75 μg/ml) doxycycline, for 9 hr. Next, 2 mM thymidine were added into the medium for 16 hr. Where needed 500 nM of auxin was added into the medium 4.5 hr before final release. Prior to harvest, all cells were pulse labeled with 10 μM EdU for 2 hr. Once released from the second thymidine block, 0 hr time point cells were harvested, and the remaining were released into fresh medium ±dox and auxin and collected at indicated time points. Samples for analysis were prepared using Click-iT EdU Alexa Fluor 488 Flow Cytometry Assay Kit following manufacturer's protocol (ThermoFisher #C10420). FxCycle Violet Stain (ThermoFisher #F10347) was used for DNA content analysis. Experiments were repeated > 3 times.

For Palbociclib cell synchronization experiments, cells were incubated with medium containing 1 μM Palbociclib (#S1116, Selleckchem) and 0.75 μg/ml doxycycline for 28 hr before first harvest. When needed auxin was added 4.5 hr prior to harvest. Another subset of cells was maintained in medium containing palbociclib and dox for another 12 hr (with or without auxin) for the second harvest.

## Mitotic index flow cytometry

TO-HCT116 and ORC2_H-2 cells were pre-treated with doxycycline for 24 hr where needed in this experiment. Cells were harvested at different time points after auxin treatment, and immediately fixed with 4% paraformaldehyde (PFA) in PBS for 15 min, mixed with 1% BSA-PBS and centrifuged and supernatant discarded. Next, cells were permeabilized with 0.5% Triton X −100 in 1% BSA-PBS for 15 min at room temperature, mixed with 1% BSA-PBS and centrifuged and supernatant discarded. Samples were incubated with anti-pH3S10 antibody (mouse monoclonal #9706, Cell Signaling) for 45 min at 37°C. Cells were then washed with 1% BSA-PBS + 0.1 % NP-40, and incubated with secondary antibody (Donkey anti-Mouse Alexa Fluor 647, #715-605-151, Jackson ImmunoResearch) for 50 min at 37°C protected from light. Finally, DNA in cells were stained with FxCycle Violet Stain (ThermoFisher) and samples analyzed by flow cytometry.

## Cell extraction and MCM2 flow cytometry

EdU pulse-labeled asynchronous TO-HCT116, ORC2_H-2, ORC2_H-5 cells with or without doxycycline and auxin treatment were harvested, washed with PBS, and processed based on the previously described protocol (*Matson et al., 2017*) with minor optimizations. In brief, for non-extracted cells, cells were fixed with 4% PFA in PBS for 15 min, and then centrifuged at 1000 xg for 7 min to remove fixation, then washed with 1% BSA-PBS and centrifuged again. Next, cells were permeabilized with 0.5% Triton X-100 in 1% BSA-PBS for 15 min and then washed with 1% BSA-PBS. For chromatin extracted cells, cells were incubated in CSK buffer (10 mM PIPES/KOH pH 6.8, 100 mM NaCl, 300 mM sucrose, 1 mM EGTA, 1 mM MgCl$_2$, 1 mM DTT) containing 0.5% Triton X-100 with protease and phosphatase inhibitors, on ice for 5 min. Cells were centrifuged, washed with 1% BSA-PBS and then fixed in 4% PFA in PBS for 15 min. After a PBS wash, samples were prepared using Click-iT EdU Alexa Fluor 488 Flow Cytometry Assay Kit manufacturer's manual (ThermoFisher #C10420), but instead of the kit's permeabilization and wash reagent, we used 1% BSA-PBS + 0.1 % NP-40 for all washing steps. Next, cells were incubated with anti-MCM2 antibody (mouse monoclonal #610700, BD Biosciences) at 37°C for 40 min protected from light. Cells were washed and then incubated with secondary antibody (Donkey anti-Mouse Alexa Fluor 647 #715-605-151 Jackson ImmunoResearch) at 37°C for 50 min, protected from light. Finally, cells were washed and stained with FxCycle Violet Stain (ThermoFisher). A total of >10,000 cells were quantified per condition. Gating for MCM negative cells was done based on unstained cell populations and secondary antibody only fluorescence controls.

## Immunofluorescence staining

TO-HCT116, ORC2_H-2, ORC2_H-4, and ORC2_H-5 cells were grown on coverslips for 48 hr with or without doxycycline and auxin treatment. Samples on coverslips were fixed in 4% PFA for 10 min at room temperature. Next, coverslips were washed for 5 min with cold PBS. Cells were then permeabilized in 0.5% Triton X-100 in PBS for 9 min. Following PBS washes, cells were blocked with 5% normal goat serum in PBS + 0.1% Tween (NGS-PBST) for 1 hr. For primary antibody incubation, antibodies were diluted in 1% NGS-PBST and incubated for overnight at 4°C. Primary antibodies used include anti-CENP-C (Mouse monoclonal #ab50974, Abcam), anti-CENP-C (Rabbit polyclonal #ABE1957, Millipore), anti-HP1α (Mouse monoclonal #MAB3584, Millipore), anti-pCHK1(S345) (rabbit polyclonal #2348, Cell Signaling), anti-p-γH2AX(S139) (rabbit monoclonal #9718, Cell Signaling), and anti-pATM(S1981) (Mouse monoclonal #ab36180, Abcam). Cells were washed with 1% NGS-PBST before incubation with secondary antibody for 1 hr at room temperature. Secondary antibodies used include Goat Anti-Mouse IgG H and L Alexa Fluor 647 (#ab150115, Abcam) and Goat Anti-Rabbit IgG H and L Alexa Fluor 488 (#ab150077, Abcam). Finally, cells were stained with 1 µg/ml DAPI and coverslips mounted with VECTASHIELD Antifade Mounting Medium (#H-1000–10, Vector Laboratories). Images were taken using a Perkin Elmer spinning disc confocal equipped with a Nikon-TiE inverted microscope using 60X objective oil lens with an Orca ER CCD camera. Images presented are maximum intensity projections of a z-stack (z = 0.3 µM).

To study the nuclear and cellular morphology HCT116 *p53$^{+/+}$*, HCT116 *p53$^{-/-}$*, *ORC1$^{-/-}$*(B14), and *ORC2$^{-/-}$* (P44) cells were grown on coverslips. Cells were fixed with 4% PFA and the method described above was followed. Primary antibody against Lamin B1 (Abcam ab16048) was used as a marker for nuclear envelope. Secondary antibody used is Goat Anti-Rabbit IgG Alexa Fluor 594 (Abcam ab150084). In addition, Phalloidin iFluor 488 (Abcam ab176753) was used to stain for cytoskeleton and DNA was detected with 1 µg/mL Hoechst dye (ThermoFisher #62249). Mounted coverslips were imaged with Perkin Elmer spinning disc confocal equipped with a Nikon-TiE inverted microscope using 40X objective lens with an Orca ER CCD camera. Images presented are single channel average intensity projections or merged multi-channel maximum intensity projections of z-stacks.

## Nuclear volume quantitation

Nuclei were fixed and stained with DRAQ5 Fluorescent Probe Solution as per the manufacturer's guidelines (ThermoFisher #62251). Images were taken using a Perkin Elmer spinning disc confocal equipped with a Nikon-TiE inverted microscope using 60X objective lens with an Orca ER CCD

camera. Images presented are maximum intensity projections of a z-stack (z = 0.3 µM). Nuclear size was analyzed with volocity software (version 6.3.1).

## Live cell microscopy

TO-HCT116 and ORC2_H-2 cells were seeded in ibidi µ-Slide 8-Well Glass Bottom, in the presence or absence of 0.75 µg/mL doxycycline for 24 hr. Next, 2 mM thymidine was added and samples incubated ±dox for 24 hr. Two hours prior to washing out thymidine, 500 nM auxin were added to the dox treated wells. Samples were then imaged beginning 4 hr after thymidine release and the timepoints reconstructed from time-lapse images using volocity software. Images were acquired approximately every 5 min on a Perkin Elmer spinning disc confocal equipped with a Nikon-TiE inverted microscope using 40X objective lens with an Orca ER CCD camera. Images presented are maximum intensity projections of a z-stack (z = 3 µM).

## Quantitative PCR

Total RNA of HCT116 $p53^{-/-}$ and $ORC2^{-/-}$ cells were extracted using Rneasy Mini Kit (Qiagen #74104) following manufacturer's handbook and quantified by Nanodrop (ThermoFisher). cDNA was made using TaqMan Reverse Transcription Reagents with either oligo(dT) or random hexamer primers (#N8080234, Applied Biosystems). Real-time quantitative PCR were performed using PowerSYBR Green PCR Master Mix (Applied biosystems #4367659) following manufacturer's protocol. Primer pairs for quantitative PCR were designed to PCR exon-exon junction (*Supplementary file 3*) and each PCR was performed as a triplicate. The delta-Ct (ΔCt) values were obtained from subtracting Actin mean Ct values from test samples. The delta-delta-Ct (ΔΔCt) value were calculated by subtracting HCT116 $p53^{-/-}$ ΔCt from $ORC2^{-/-}$ ΔCt for each primer pair individually. Data is represented as Log2 Fold change (FC) for each primer pair in $ORC2^{-/-}$ cells compared to HCT116 $p53^{-/-}$ cells.

## Transmission electron microscopy

HCT116 $p53^{-/-}$ and $ORC1^{-/-}$ cells were pelleted and resuspended in 1 mL of 2.5% glutaraldehyde in 0.1 M sodium cacodylate solution (pH 7.4) overnight at 4˚C. Fixative was removed, and in each step ~ 200 µl of the solution was left in the tubes. Pellet was washed with 0.1 M sodium cacodylate buffer. Next, 4% low melting agarose solution was added and the tubes centrifuged immediately at 1000 x g for 10 min at 30˚C, and transferred directly on ice for 20 min to solidify the agarose. Agarose was washed twice with 0.1 M cacodylate buffer. Next, 1% osmium tetraoxide (OsO4) solution was added and left undisturbed for 1 hr followed by three 0.1 M cacodylate buffer washes. Samples were then serially dehydrated using increasing amounts of ethanol (50, 60, 70, 80, 90, 95, and 100%, respectively). Finally, samples were embedded in 812 Embed resin and sectioned into 60–90 µm sections using Ultramicrotome. Hitachi H-7000 Transmission Electron Microscopy was used to image the sample.

## Copy number variation analyses by SMASH

Copy number profiles were generated from input DNA using the SMASH sequencing protocol and analysis pipeline as described previously (*Wang et al., 2016*). Briefly, total cellular genomic DNA was isolated from HCT116 $p53^{+/+}$, HCT116 $p53^{-/-}$, $ORC1^{-/-}$, and $ORC2^{-/-}$ cell lines using QIAamp mini kit (Qiagen, 51104). Approximately 500 ng genomic DNA was enzymatically fragmented using dsDNA fragmentase (NEB, M0348L). Following end repair, fragments were joined to create chimeric fragments of DNA suitable for creating NGS libraries (300–700 bp). The fragment size selection was done with Agencourt AMPure XP beads (Beckman Coulter, Cat. No. A63881). Illumina-compatible NEBNext Multiplex Dual Index Primer Pairs and adapters (New England Biolabs, Cat. No. E6440S) were ligated to the selected chimeric DNA fragments. These barcoded DNA fragments were then sequenced using an Illumina 300cycle MiSeqv2 kit on a MiSeq platform.

The SMASH analysis pipeline searches for Maximal Unique Matches (MUMs) to the human genome in all read pairs using a suffix array. These MUMs were then filtered to exclude short matches below 20 bp, matches with less than 4 bp of excess unique sequence, and matches on read two that are within 1000 bases of the genome coordinate of matches from read 1. The resulting 3–4 on average kept matches per read pair are then added to pre-computed empirically sized bins spanning the genome to generate a raw copy number profile. Regions with identical copy are expected

to yield similar bin counts using these empirical bins. This profile is then corrected to remove GC content effects by normalizing counts based on LOWESS smoothing of count vs. GC content data in each bin. Final copy number profiles are normalized so that the autosome has an average copy number of 2. Plots were generated with G-Graph MUMdex software - MUMdex Genome Alignment Anal. Softw. (https://mumdex.com/) (*Andrews et al., 2016*).

## Oxford nanopore technologies (ONT) long read sequencing and analysis

High-molecular-weight DNA was isolated using the MagAttract kit (Qiagen # 67563). DNA was sheared to 50 kb via Megarupter (diagenode). The quality of the DNA was then assessed on a Femtopulse (Agilent) to ensure DNA fragments were > 40 kb on average. After shearing, the DNA was size selected with an SRE kit (Circulomics) to reduce the fragments size to < 20 kb. After size selection, the DNA underwent a-tailing and damage repair followed by ligation to sequencing specific adapters.

Half of the prepared library was mixed with library loading beads and motor protein and then loaded on to an ONT PromothION PROM-0002 flow-cell and allowed to sequence for 24 hr. After 24 hr, the flow-cell was treated with DNase to remove stalled DNA followed by a buffer flush. The second half of the library was then loaded and allowed to sequencing for 36 hr. The DNA was base called via Guppy 3.2 in High accuracy mode.

Long reads were aligned to the reference human genome using NGMLR (https://github.com/phil-res/ngmlr) and structural variants were identified using Sniffles (https://github.com/fritzsedlazeck/Sniffles) (*Sedlazeck et al., 2018*). The alignments and structural variants were then visualized using IGV (https://igv.org/).

## Acknowledgements

We thank Dr. Leemor Joshua-Tor for comments on the manuscript and Dr. Anindya Dutta for providing cell lines. We also thank Jennifer Shapp for technical assistance.

## Additional information

### Competing interests

Bruce Stillman: Reviewing editor, *eLife*. The other authors declare that no competing interests exist.

### Funding

| Funder | Grant reference number | Author |
|---|---|---|
| National Cancer Institute | P01-CA13106 | Bruce Stillman Christopher R Vakoc W Richard McCombie |
| National Cancer Institute | P50-CA045508 | Bruce Stillman |
| National Science Foundation | DBI-1627442 | Michael C Schatz |

The funders had no role in study design, data collection and interpretation, or the decision to submit the work for publication.

### Author contributions

Hsiang-Chen Chou, Kuhulika Bhalla, Conceptualization, Data curation, Formal analysis, Validation, Visualization, Methodology, Writing - original draft; Osama EL Demerdesh, Sergey Aganezov, Data curation, Formal analysis, Writing - review and editing; Olaf Klingbeil, Habeeb Alsudani, Kenneth Chang, Methodology, Writing - review and editing; Kaarina Hanington, Methodology; Peter Andrews, Formal analysis, Methodology, Writing - review and editing; Christopher R Vakoc, Conceptualization, Supervision, Funding acquisition, Methodology, Writing - review and editing; Michael C Schatz, W Richard McCombie, Data curation, Formal analysis, Funding acquisition, Methodology, Writing - review and editing; Bruce Stillman, Conceptualization, Data curation, Formal analysis,

Supervision, Funding acquisition, Investigation, Writing - original draft, Project administration, Writing - review and editing

### Author ORCIDs
Christopher R Vakoc ![iD] http://orcid.org/0000-0002-1158-7180
Bruce Stillman ![iD] https://orcid.org/0000-0002-9453-4091

### Decision letter and Author response
Decision letter https://doi.org/10.7554/eLife.61797.sa1
Author response https://doi.org/10.7554/eLife.61797.sa2

## Additional files

### Supplementary files
• Supplementary file 1. The sequences of all guide RNAs used for gene editing, including those directed to ORC1-6 and CDC6 as well as positive and negative guides for the tiling CRISPR screens.

• Supplementary file 2. Sequence of Barcode primers used for Next Gene Sequencing analysis in tiling CRISPR screens.

• Supplementary file 3. Primers used for exon analysis qPCR of the *ORC2* gene cDNAs from various cell lines.

• Transparent reporting form

### Data availability
DNA sequencing data including Tiling-sgRNA CRISPR screen, CNV analysis by SMASH and ONT Nanopore long-read sequencing are available in the Dryad database.

The following dataset was generated:

| Author(s) | Year | Dataset title | Dataset URL | Database and Identifier |
|---|---|---|---|---|
| Chou H-C, Bhalla K, Demerdesh OE, Klingbeil O, Hanington K, Aganezov S, Andrews P, Alsudani H, Chang K, Vakoc C R, Schatz M, McCombie WR, Stillman B | 2020 | Data from: The human Origin Recognition Complex is essential for pre-RC assembly, mitosis and maintenance of nuclear structure. | https://doi.org/10.5061/dryad.59zw3r25f | Dryad Digital Repository, 10.5061/dryad.59zw3r25f |

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
