## [Decision Letter]

**Acceptance summary:**

The mechanisms that allow for a very limited capacity human cells, both oncogenically transformed or not, have to replicate and divide without ORC1 or ORC 2 remains mysterious. However, establishing the essential requirements for wild type replication for the complex is important as are the phenotypes associated with loss of a functional ORC.

**Decision letter after peer review:**

Thank you for submitting your article "The human Origin Recognition Complex is essential for pre-RC assembly, mitosis and maintenance of nuclear structure" for consideration by *eLife*. Your article has been reviewed by three peer reviewers, and the evaluation has been overseen by Reviewing Editor Michael Botchan and Kevin Struhl as the Senior Editor. The following individual involved in review of your submission has agreed to reveal their identity: Matthew W Parker (Reviewer #2).

The reviewers have discussed the reviews with one another and the Reviewing Editor has drafted this decision to help you prepare a revised submission.

Overall all three reviewers felt the work establishes that the ORC subunits and Cdc6 are essential for DNA replication in transformed human cell lines and this result clarifies in important ways how Shibata et al. in their 2016 *eLife* paper may have been misled in regard to ORC2 for sure. However, some adaption may have occurred in ORC 1 null cell lines at some time prior to or at the time of mutation. This null ORC1 line does replicate for roughly 30 generations and contrasts starkly to what happens with the ORC2 nulls. Why? No real discussion is presented and it may it be that Shibata was on to something for cancer cells? The reviewing editor wonders if a repair dependent process for an ORC1 null line may let cells with subsequent defective DNA mutations arise, with the mutations then ultimately leading to cell death or senescence? Investigating the loading of the MCMs may help clarify these questions.

Experiments are suggested by the reviewers in conversation and first by reviewer #2. Do the authors have an idea of what level of replication is occurring in the Orc1 null cells during propagation? If not, they should implement their FACS/Edu experiment to assess this; the FACS/MCM loading assay might also prove informative. The reference to Asano and Park (PNAS 2008; reviewer #3) does set a precedent for the point that certain types of DNA replication may occur in special cell types with "undetectable" levels of ORC1; e.g., polytene cell replication occurring in salivary glands (although in a subsequent paper certain ORC1 mutations were reported as female sterile and defects were found in follicle cell amplification). However, in tissue with very low levels of ORC 1 from maternal stores, not detectable by protein blot, some MCM loading may still occur, especially in heterogeneous tissue where the majority of protein comes from post replicating cells. We feel the cell-type specific points with these established human cell lines are essential for publication, given the major implications for cancer, which are still controversial even given that ORC 1 null mutations are lethal for both *Drosophila melanogaster* and mice.

The reviewers also point out that the present work does not show directly a role for ORC subunits in non-DNA replication functions. The manuscript should clarify what has been done elsewhere to indicate either a direct ORC role in non-replication functions or simply hedge conclusions. We have left each of the reviews for your perusal as many suggestions will help in the revised manuscript. However, again the essential points outlined above will be necessary for *eLife* publication.

Reviewer #1:

This manuscript by Chou, Bhalla et al., directly addresses a recent controversy in the replication field as to whether ORC1 and ORC2 are essential genes (Shibata et al., 2016). Using a sgRNA-tiling CRISPR screen specifically targeting all ORC subunits and Cdc6, the authors find all ORC genes and Cdc6 to be essential. The authors then create a cell line that allows for the rapid depletion of ORC2 and characterize the phenotypes upon depletion. They describe several molecular and cellular phenotypes associated with ORC2 depletion. Finally, they reveal that the previously published ORC2 mutant is not a true null mutant.

Overall, I find the authors data showing that ORC and Cdc6 genes are essential compelling, which agrees with a previous published CRISPR-based screen of cell cycle regulated genes (McKinley and Cheeseman, 2017). The phenotypes described upon depletion of ORC2 are also compelling, however, many of these phenotypes have been described in numerous publications with siRNA-mediated depletion of ORC. Furthermore, the authors fall short in determining which phenotypes are due to loss of specific functions of ORC or secondary phenotypes due to DNA damage accumulation in ORC-depleted cells. This could be addressed by providing qualifying statements in the text.

1) This paper has many typos and grammatical mistakes. Especially in the Materials and methods section. It would be nice to have a native English speaker read the entire manuscript – including the Materials and methods section.

2)Figure 2F-H: Why are the ORC2-4 and ORC2-4 sgRNAs not complemented by the mAID-ORC2 construct? Is the construct not resistant to those gRNAs? Or, is the result of off-target effects?

3) “ORC1 and Cdc6 levels remain unchanged”. This is not true based on the data in Figure 3G. You should quantify the data in Figure 3G.

4) The authors should quantify their IF images (e.g. # foci/cell). Figures 3H, I, 4C.

5) Figure 4: Is this a direct effect of ORC2 depletion? The authors could look at heterochromatin compaction in synchronized cells prior to S phase.

6) Subsection “ORC2 is essential for initiation of DNA replication”: Figure 5: To understand the essential function of ORC it would be nice to know if ORC was degraded prior to release. This experiment would be interpreted very differently if ORC was completely degraded and the first cell cycle proceeded normally vs. if ORC wasn't fully degraded upon release. I'm not sure if the defects in the second cell cycle are a reflection of lack of ORC or genome instability in the first cell cycle. Therefore, I don't think the authors can make the conclusions in the aforementioned subsection without knowing this.

7) Subsection “An MCM complex loading and pre-RC assembly defect in ORC2 depleted cells”, Figure 6: It would be important to know the level of detection in this assay. The authors say there are no MCMs bound upon ORC2 depletion. But the “negative” cells could have ~10% of WT levels of MCMs. It is known that cells can replicate normally with drastically reduced MCM levels.

8) Figure 7: Is this a direct effect of ORC2 depletion or an indirect effect of checkpoint activation?

9) Discussion, end of first paragraph: I don't think this statement is true given that some rapidly evolving yeast species lack many DNA replication and repair proteins (Steenwyk et al. 2019 PLoS Biology (https://doi.org/10.1371/journal.pbio.3000255).

Reviewer #2:

In this manuscript from Chou et al. the authors utilize tiling sgRNA CRISPR screens and experiments in previously published Orc null lines to confirm the essential role of origin recognition complex subunits in cell proliferation of multiple human cancer cell lines. The significance of the work lies primarily in rebutting recent claims published in *eLife* that certain components of the human origin recognition complex are dispensable for proliferation (Shibata et al., 2016), and so represents an important contribution to this discussion. The paper is well written, the approach implemented is thorough and the major conclusions substantiated by the data. Additionally, some new biology is gleamed regarding a general role of ORC in maintaining nuclear architecture that opens the door for future investigation. With one additional experiment to confirm their Orc2 knockout and some added discussion/experimentation regarding the previous Orc1 null line (both points detailed below) the paper will be suitable for publication in *eLife*.

1) The paper is largely framed by the previous findings from Shibata et al. that Orc1 and Orc2 are dispensable for proliferation of human cancer cell lines. While the authors clearly show that Orc2 is essential, their data regarding Orc1 is more ambiguous. Indeed, they observe, consistent with Shibata et al., that the Orc1 null line is a true knockout and yet this cell line is still capable of proliferation. This indicates that some level of DNA replication is occurring, as incomplete replication would quickly halt proliferation (as indicated by their ORC2 knockout line which stops proliferating after a single round of division). Do the authors have an idea of what level of replication is occurring in the Orc1 null cells? If not, they should implement their FACS/Edu experiment to assess this; the FACS/Mcm loading assay might also prove informative. And if replication/Mcm loading is occurring, do the authors agree with the Shibata conclusion, that Orc1 is not absolutely essential for replication in human cancer cells, at least for the short term (fewer than 30 generations)? How might cancer cells cope with this from a mechanistic standpoint? The authors should expand their Discussion to include these points.

2) The authors' ORC2_1 sgRNA lines are clearly functional knockouts, as indicated by the inability of these cells to proliferate when mAID-ORC2 is depleted. This approach, which included an inducible degron and select resistance to sgRNAs, was well-designed and meticulously executed. However, it is not clear whether Orc2 protein is completely lost in these cells. They state that "no truncated form of protein was detected". However, their western blot in Figure 3A shows a band migrating slightly faster than the endogenous Orc2 and, given the deletion size of approximately 50 amino acids (or 5 kDa), this species seems to be approximately the correct size of a C-terminal truncated protein. The authors only state that this is a "nonspecific smaller band" but do not provide any data to support this conclusion. Do the authors have additional evidence to add? If not, the authors should use their Orc3 IP experiment (Figure 8—figure supplement 1), for which they have all reagents in hand and which was used to verify that the Shibata Orc2^-/-^ is not a true knockout, to confirm in their Orc2_H-2, H-4 and H-5 lines that the faster migrating band is, in fact, a non-specific protein that does not pull down with Orc3 (with positive controls run simultaneously).

Reviewer #3:

The manuscript by Stillman and colleagues describes a detailed analysis of the consequences of eliminating individual subunits of the human DNA replication initiator protein, ORC. This has been a controversial area as some authors (notably Shibata et al) have suggested that ORC1 and ORC2 are not essential, while others (notably McKinley and Cheeseman) have suggested that they are essential. In this manuscript the authors use a combination of high density CRISPR mutagenesis and cell lines that allow rapid depletion to come to the conclusion that all of the ORC subunits and Cdc6 are essential. In addition, the authors perform extensive analysis of strains that allow rapid ORC2 depletion, observing defects in DNA replication, chromosome condensation, and mitosis. Finally, they analyze the previously described cell lines that grow in the absence of ORC1 or ORC2 to address how they are still able to grow.

In the first part of the paper the authors use a combination of CRISPR targeting and competitive cell growth as a mechanism of assessing the importance of each ORC gene and Cdc6. For each gene they created a series of guide RNAs that spanned the full coding region of each gene and then asked how cells treated with the guide RNA grew relative to control untreated cells. By comparison of the growth differences observed to control guide RNAs that, the authors conclude that all six ORC genes and Cdc6 are essential. What is less clear is whether the authors can conclude that the site of targeting reveals the importance of that region. Although the authors state that targeting sites of conserved regions have more impact than non-conserved regions, there are many instances when this correlation is not observed. In addition, when there are strong effects in non-conserved regions (e.g. the IDRs of ORC1 and ORC2) the authors conclude that they are essential regions of the protein. It is not clear that the differing effects could just as likely be due to the frequency of making deletions and frame shifts at different regions rather than their conservation(for example, if the local DNA sequences have micro-homologies that lend themselves to deletions). If the authors want to make such conclusions they need to provide a more quantitative analysis than what is shown in the current manuscript (e.g. show a correlation between the LFC and the extent of conservation of the targeted region – i.e. the 20 aa on either side of the CRISPR cut site).

Much of the remaining data concerns the analysis of cell lines in which ORC2 can be rapidly removed from cells by regulated degradation. They show that removal of ORC2 leads to rapid loss of proliferation and cell cycle arrest. Using markers for the DNA damage checkpoint they also show that this checkpoint is activated in these cells. They further find that depletion of ORC2 results in aberrant nuclear morphology and decondensation of centromeric heterochromatin. Not surprisingly, they also see defects in replication initiation. This is not particularly well demonstrated by the data in Figure 5 as all the defects observed could be due to difficulty in elongation. However, the data in Figure 6 addressing MCM2 association with chromatin are more compelling and are certainly in line with the known functions of ORC. Finally, the authors observe ORC2 depletion also results in defects in centromeric condensation, nuclear morphology, and entry into and completion of mitosis.

The last section of the paper looks into the previously identified cell lines "lacking" ORC1 or ORC2 and comes to different conclusions regarding the two cell lines. For the ORC2^-/-^ line, the conclusion is simple. It is clear that there is still functional ORC2 present. In the case of the ORC1^-/-^ cell line, this does not seem to be the case. Instead, the authors focus on the ORC1^-/-^ cell line characteristics. They observe that it has limited proliferation capacity of 30 generations or less and that it has significant morphological and cell division defects. Nevertheless, the data seems to show that ORC1 is not required for successful cell division and DNA replication (similar to what was reported by Shibata et al.). The authors should do a better job of saying that they agree with the previous studies and discuss more about how this could be the case instead of only focusing on the ORC2^-/-^ strain not being a deletion. For example, there are other studies in *Drosophila* also observing ORC1 as being non-essential for DNA replication (Park, S.Y., and Asano, M. (2008). The origin recognition complex is dispensable for endoreplication in *Drosophila*. Proceedings of the National Academy of Sciences 105, 12343-12348.). Certainly, it is true that these cells are unhappy and at a severe competitive advantage, but they are replicating their DNA and proliferating in the absence of ORC1 and this observation remains unexpected and worthy of discussion by the authors.

Overall, this paper provides strong evidence that ORC is an essential DNA replication protein. The authors also identify a myriad of defects of ORC depletion beyond DNA replication (some of which have been observed previously by the Stillman using less refined ORC depletion methods). What is not clear based on the data, is whether the defects in chromatin condensation, nuclear morphology, and mitosis are due to direct effects of ORC on these events versus indirect defects of the incomplete DNA replication that occurs as a consequence of ORC depletion. This alternative possibility should be discussed more directly by the authors as there is nothing in the paper to suggest that the effects are direct (e.g. experiments removing ORC function in G1 or M phase arrested cells).

---

## [Author Response]

Overall all three reviewers felt the work establishes that the ORC subunits and Cdc6 are essential for DNA replication in transformed human cell lines and this result clarifies in important ways how Shibata et al. in their 2016 eLife paper may have been misled in regard to ORC2 for sure. However, some adaption may have occurred in ORC 1 null cell lines at some time prior to or at the time of mutation. This null ORC1 line does replicate for roughly 30 generations and contrasts starkly to what happens with the ORC2 nulls. Why? No real discussion is presented and it may it be that Shibata was on to something for cancer cells? The reviewing editor wonders if a repair dependent process for an ORC1 null line may let cells with subsequent defective DNA mutations arise, with the mutations then ultimately leading to cell death or senescence? Investigating the loading of the MCMs may help clarify these questions.

We have added new discussion in the paper that addresses the possible mechanism for why the *ORC1* depleted cells proliferate, although we want to emphasize that these cells hardly proliferate and their life span is very limited, so ORC is essential in both transformed and diploid cells. The *ORC2* cells reported by Shibata et al. are not null, they are defective in full length ORC2, but have a complementing truncated ORC2 protein that is sensitive to CRISPR-Cas9 mediated inhibition.

Experiments are suggested by the reviewers in conversation and first by reviewer #2. Do the authors have an idea of what level of replication is occurring in the Orc1 null cells during propagation? If not, they should implement their FACS/Edu experiment to assess this; the FACS/MCM loading assay might also prove informative. The reference to Asano and Park (PNAS 2008; reviewer #3) does set a precedent for the point that certain types of DNA replication may occur in special cell types with "undetectable" levels of ORC1 ; e.g., polytene cell replication occurring in salivary glands (although in a subsequent paper certain ORC1 mutations were reported as female sterile and defects were found in follicle cell amplification). However, in tissue with very low levels of ORC 1 from maternal stores, not detectable by protein blot, some MCM loading may still occur, especially in heterogeneous tissue where the majority of protein comes from post replicating cells. We feel the cell-type specific points with these established human cell lines are essential for publication, given the major implications for cancer, which are still controversial even given that ORC 1 null mutations are lethal for both *Drosophila melanogaster* and mice.

We show using CRISPR-Cas9 mediated gRNA tiling that ORC1 is essential in both cancer cells and in human diploid cells and we have added cell proliferation data to show that the *ORC1^-/-^* depleted cells have a very slow doubling rate and can only divide for a limited number of generations, and have many cells with a highly abnormal nuclear morphology.

The reviewers also point out that the present work does not show directly a role for ORC subunits in non-DNA replication functions. The manuscript should clarify what has been done elsewhere to indicate either a direct ORC role in non-replication functions or simply hedge conclusions. We have left each of the reviews for your perusal as many suggestions will help in the revised manuscript. However, again the essential points outlined above will be necessary for eLife publication.Reviewer #1:This manuscript by Chou, Bhalla et al., directly addresses a recent controversy in the replication field as to whether ORC1 and ORC2 are essential genes (Shibata et al., 2016). Using a sgRNA-tiling CRISPR screen specifically targeting all ORC subunits and Cdc6, the authors find all ORC genes and Cdc6 to be essential. The authors then create a cell line that allows for the rapid depletion of ORC2 and characterize the phenotypes upon depletion. They describe several molecular and cellular phenotypes associated with ORC2 depletion. Finally, they reveal that the previously published ORC2 mutant is not a true null mutant.Overall, I find the authors data showing that ORC and Cdc6 genes are essential compelling, which agrees with a previous published CRISPR-based screen of cell cycle regulated genes (McKinley and Cheeseman, 2017). The phenotypes described upon depletion of ORC2 are also compelling, however, many of these phenotypes have been described in numerous publications with siRNA-mediated depletion of ORC. Furthermore, the authors fall short in determining which phenotypes are due to loss of specific functions of ORC or secondary phenotypes due to DNA damage accumulation in ORC-depleted cells. This could be addressed by providing qualifying statements in the text.

We thank the reviewer and have made changes to the Discussion to highlight defects in context. We have also added new data that clarifies the change in nuclear morphology upon ORC2 depletion and when this occurs during the cell division cycle.

1) This paper has many typos and grammatical mistakes. Especially in the Materials and methods section. It would be nice to have a native English speaker read the entire manuscript – including the Materials and methods section.

We have edited the manuscript.

2) Figure 2F-H: Why are the ORC2-4 and ORC2-4 sgRNAs not complemented by the mAID-ORC2 construct? Is the construct not resistant to those gRNAs? Or, is the result of off-target effects?

The mAID-ORC2^gr^ construct was made specifically to be resistant to ORC2-1 and ORC2-2 sgRNAs (Figure 2D). It is not expected that the ORC2-3 and ORC2-4 gRNAs will be complemented since they still cut the transgene. The fact that we could see specificity for the gRNAs in the complementation argues strongly against off target effects.

3) “ORC1 and Cdc6 levels remain unchanged”. This is not true based on the data in Figure 3G. You should quantify the data in Figure 3G.

We’ve quantified the levels of ORC1 and CDC6 in Figure 3G in new Figure 3—figure supplement 3B and C.

4) The authors should quantify their IF images (e.g. # foci/cell). Figures 3H, I, 4C.

We’ve quantified the IF images. Quantification of Figure 3H and 3I are now in Figure 3—figure supplement 3D and E, respectively. Quantification of Figure 4C is now in Figure 4—figure supplement 1.

5) Figure 4: Is this a direct effect of ORC2 depletion? The authors could look at heterochromatin compaction in synchronized cells prior to S phase.

To address this question, Cells were synchronized in G1 phase followed by ORC2 depletion and the result is shown in new Figure 4—figure supplement 3A and B. While the cells stayed in G1, we harvested cells at two different time points after auxin treatment. The nuclear volume was unchanged. CENP-C staining was also unchanged (data not shown). We suggested that this abnormal heterochromatin phenotype we observed after ORC depletion in asynchronous cell is due to cells lacking ORC2 during S phase that affect sister chromatid cohesion loading (Zheng et al., 2018). The centromeric region showed decompaction in late S/ G2 phase and we confirmed this prediction by demonstrating re-arrangement of heterochromatin protein HP1α concomitant with an increase in CENP-C staining, providing evidence of a global nuclear decondensation upon ORC2 depletion (new Figure 4—figure supplement 2) We compared the HP1α staining in HCT116 cells and HeLa cells since we have previously studied the localization of HP1α in cells depleted of ORC subunits and HeLa, unlike HCT116 cells, have clustered heterochromatin foci.

6) Subsection “ORC2 is essential for initiation of DNA replication”: Figure 5: To understand the essential function of ORC it would be nice to know if ORC was degraded prior to release. This experiment would be interpreted very differently if ORC was completely degraded and the first cell cycle proceeded normally vs. if ORC wasn't fully degraded upon release. I'm not sure if the defects in the second cell cycle are a reflection of lack of ORC or genome instability in the first cell cycle. Therefore, I don't think the authors can make the conclusions in the aforementioned subsection without knowing this.

We have shown the depletion of mAID-ORC2^gr^ in a new Figure 5—figure supplement 1A. The mAID-ORC2^gr^ was depleted upon release from a second thymidine block in the auxin-treated ORC2_H-2 cells group. The depletion persists until the last collection at 48 hr. (Lane 23-28). We proposed that although mAID-ORC2^gr^ was depleted upon release, the pre-RC was already formed during the second thymidine incubation. Thus the first S phase is normal. However, during the second cell cycle, ORC2 is entirely lost from the cells since the previous S, G2 and M phases, and therefore cells begin to show cell cycle defects consistent with a loss of pre-RC assembly.

7) Subsection “An MCM complex loading and pre-RC assembly defect in ORC2 depleted cells” Figure 6: It would be important to know the level of detection in this assay. The authors say there are no MCMs bound upon ORC2 depletion. But the “negative” cells could have ~10% of WT levels of MCMs. It is known that cells can replicate normally with drastically reduced MCM levels.

We changed the layout plot in a new Figure 6 to show the MCM2 detection level in each experimental group. As shown in the new data, within the G1 population, ORC2_H-2 cells have only 4.6 % MCM2^DNA^ pos., and the MCM2^DNA^ neg. population is shifted downward, suggesting that “negative” cells do not retain MCM2.

8) Figure 7: Is this a direct effect of ORC2 depletion or an indirect effect of checkpoint activation?

It is hard to tell whether it’s an indirect effect of checkpoint activation. While most cells have checkpoint activation, only a very small percentage of the cells enter mitosis to exhibit the abnormal phenotype. ORC2 localizes at the kinetochore during M phase, and we suggest that those mitotic cells lacking ORC2 cannot properly align and segregate sister chromatids during M phase. But we do not know if ORC2 directly controls or is essential for recruiting other factors sometime during or before M phase for sister chromatid alignment.

9) Discussion, end of first paragraph: I don't think this statement is true given that some rapidly evolving yeast species lack many DNA replication and repair proteins (Steenwyk et al. 2019 PLoS Biology (https://doi.org/10.1371/journal.pbio.3000255).

This sentence has been edited.

Reviewer #2:In this manuscript from Chou et al. the authors utilize tiling sgRNA CRISPR screens and experiments in previously published Orc null lines to confirm the essential role of origin recognition complex subunits in cell proliferation of multiple human cancer cell lines. The significance of the work lies primarily in rebutting recent claims published in eLife that certain components of the human origin recognition complex are dispensable for proliferation (Shibata et al., 2016), and so represents an important contribution to this discussion. The paper is well written, the approach implemented is thorough and the major conclusions substantiated by the data. Additionally, some new biology is gleamed regarding a general role of ORC in maintaining nuclear architecture that opens the door for future investigation. With one additional experiment to confirm their Orc2 knockout and some added discussion/experimentation regarding the previous Orc1 null line (both points detailed below) the paper will be suitable for publication in eLife.1) The paper is largely framed by the previous findings from Shibata et al. that Orc1 and Orc2 are dispensable for proliferation of human cancer cell lines. While the authors clearly show that Orc2 is essential, their data regarding Orc1 is more ambiguous. Indeed, they observe, consistent with Shibata et al., that the Orc1 null line is a true knockout and yet this cell line is still capable of proliferation. This indicates that some level of DNA replication is occurring, as incomplete replication would quickly halt proliferation (as indicated by their ORC2 knockout line which stops proliferating after a single round of division). Do the authors have an idea of what level of replication is occurring in the Orc1 null cells? If not, they should implement their FACS/Edu experiment to assess this; the FACS/Mcm loading assay might also prove informative. And if replication/Mcm loading is occurring, do the authors agree with the Shibata conclusion, that Orc1 is not absolutely essential for replication in human cancer cells, at least for the short term (fewer than 30 generations)? How might cancer cells cope with this from a mechanistic standpoint? The authors should expand their Discussion to include these points.

We thank the reviewer for the comments, and we have now included additional data on the ORC1 depleted cells showing that they have a cell cycle division half-life of 96hrs, compared to the parental cells that divide every 24 hrs. Furthermore, the cells cannot be maintained for more than 20-30 generation, showing that ORC1 is essential. The cells over-express CDC6 which forms a complex with the ORC subunits, and it is possible that the very compromised cell proliferation observed in the absence of ORC1 is due to compensatory over-expression of CDC6. We also show that the ORC1 depleted cells yield large number of cells with a very abnormal nuclear morphology.

2) The authors' ORC2_1 sgRNA lines are clearly functional knockouts, as indicated by the inability of these cells to proliferate when mAID-ORC2 is depleted. This approach, which included an inducible degron and select resistance to sgRNAs, was well-designed and meticulously executed. However, it is not clear whether Orc2 protein is completely lost in these cells. They state that "no truncated form of protein was detected". However, their western blot in Figure 3A shows a band migrating slightly faster than the endogenous Orc2 and, given the deletion size of approximately 50 amino acids (or 5 kDa), this species seems to be approximately the correct size of a C-terminal truncated protein. The authors only state that this is a "nonspecific smaller band" but do not provide any data to support this conclusion. Do the authors have additional evidence to add? If not, the authors should use their Orc3 IP experiment (Figure 8—figure supplement 1), for which they have all reagents in hand and which was used to verify that the Shibata Orc2^-/-^ is not a true knockout, to confirm in their Orc2_H-2, H-4 and H-5 lines that the faster migrating band is, in fact, a non-specific protein that does not pull down with Orc3 (with positive controls run simultaneously).

We’ve shown the nonspecific smaller band does not co-IP with anti-ORC3 antibody in a new Figure 3—figure supplement 1C. Thus, we conclude that this band is not able to function like an ORC2 protein that binds to ORC3.

Reviewer #3:The manuscript by Stillman and colleagues describes a detailed analysis of the consequences of eliminating individual subunits of the human DNA replication initiator protein, ORC. This has been a controversial area as some authors (notably Shibata et al) have suggested that ORC1 and ORC2 are not essential, while others (notably McKinley and Cheeseman) have suggested that they are essential. In this manuscript the authors use a combination of high density CRISPR mutagenesis and cell lines that allow rapid depletion to come to the conclusion that all of the ORC subunits and Cdc6 are essential. In addition, the authors perform extensive analysis of strains that allow rapid ORC2 depletion, observing defects in DNA replication, chromosome condensation, and mitosis. Finally, they analyze the previously described cell lines that grow in the absence of ORC1 or ORC2 to address how they are still able to grow.In the first part of the paper the authors use a combination of CRISPR targeting and competitive cell growth as a mechanism of assessing the importance of each ORC gene and Cdc6. For each gene they created a series of guide RNAs that spanned the full coding region of each gene and then asked how cells treated with the guide RNA grew relative to control untreated cells. By comparison of the growth differences observed to control guide RNAs that, the authors conclude that all six ORC genes and Cdc6 are essential. What is less clear is whether the authors can conclude that the site of targeting reveals the importance of that region. Although the authors state that targeting sites of conserved regions have more impact than non-conserved regions, there are many instances when this correlation is not observed. In addition, when there are strong effects in non-conserved regions (e.g. the IDRs of ORC1 and ORC2) the authors conclude that they are essential regions of the protein. It is not clear that the differing effects could just as likely be due to the frequency of making deletions and frame shifts at different regions rather than their conservation(for example, if the local DNA sequences have micro-homologies that lend themselves to deletions). If the authors want to make such conclusions they need to provide a more quantitative analysis than what is shown in the current manuscript (e.g. show a correlation between the LFC and the extent of conservation of the targeted region – i.e. the 20 aa on either side of the CRISPR cut site).Much of the remaining data concerns the analysis of cell lines in which ORC2 can be rapidly removed from cells by regulated degradation. They show that removal of ORC2 leads to rapid loss of proliferation and cell cycle arrest. Using markers for the DNA damage checkpoint they also show that this checkpoint is activated in these cells. They further find that depletion of ORC2 results in aberrant nuclear morphology and decondensation of centromeric heterochromatin. Not surprisingly, they also see defects in replication initiation. This is not particularly well demonstrated by the data in Figure 5 as all the defects observed could be due to difficulty in elongation. However, the data in Figure 6 addressing MCM2 association with chromatin are more compelling and are certainly in line with the known functions of ORC. Finally, the authors observe ORC2 depletion also results in defects in centromeric condensation, nuclear morphology, and entry into and completion of mitosis.The last section of the paper looks into the previously identified cell lines "lacking" ORC1 or ORC2 and comes to different conclusions regarding the two cell lines. For the ORC2^-/-^ line, the conclusion is simple. It is clear that there is still functional ORC2 present. In the case of the ORC1^-/-^ cell line, this does not seem to be the case. Instead, the authors focus on the ORC1^-/-^ cell line characteristics. They observe that it has limited proliferation capacity of 30 generations or less and that it has significant morphological and cell division defects. Nevertheless, the data seems to show that ORC1 is not required for successful cell division and DNA replication (similar to what was reported by Shibata et al.). The authors should do a better job of saying that they agree with the previous studies and discuss more about how this could be the case instead of only focusing on the ORC2^-/-^ strain not being a deletion. For example, there are other studies in *Drosophila* also observing ORC1 as being non-essential for DNA replication (Park, S.Y., and Asano, M. (2008). The origin recognition complex is dispensable for endoreplication in *Drosophila*. Proceedings of the National Academy of Sciences 105, 12343-12348.). Certainly, it is true that these cells are unhappy and at a severe competitive advantage, but they are replicating their DNA and proliferating in the absence of ORC1 and this observation remains unexpected and worthy of discussion by the authors.Overall, this paper provides strong evidence that ORC is an essential DNA replication protein. The authors also identify a myriad of defects of ORC depletion beyond DNA replication (some of which have been observed previously by the Stillman using less refined ORC depletion methods). What is not clear based on the data, is whether the defects in chromatin condensation, nuclear morphology, and mitosis are due to direct effects of ORC on these events versus indirect defects of the incomplete DNA replication that occurs as a consequence of ORC depletion. This alternative possibility should be discussed more directly by the authors as there is nothing in the paper to suggest that the effects are direct (e.g. experiments removing ORC function in G1 or M phase arrested cells).

We thank the reviewer for this analysis and for suggestions. We have analyzed the tiling sgRNA data in greater detail and have been able to show that indeed the cohort of sgRNAs that target highly conserved annotated domains display a significantly higher depletion phenotype compared to those targeting non-annotated domain regions. Furthermore, to be sure of the regions that are not part of annotated domains but seem to be hypersensitive to CRISPR-knockout we have run the Protiler analysis pipeline which denoises the data to rule out false-positives arising due to inactive sgRNA or additive effects. The new analysis has been included in entirely new Figure 2 supplements and the essential regions of the proteins are consistent between cancer and normal cells.

Additionally, previous studies that have characterized CRISPR cut site indels/mutations have reported that reproducible DNA repair patterns are not explained by microhomologies (MH). While MH lends itself to deletions, their frequency is limited and if they occur, deletions only happen when the MH is extremely close to the protospacer adjacent motif. Furthermore, an intact microhomology mediated end-joining pathway is required for this phenomenon which is absent from HCT116 cell lines – and can be beneficial in identifying novel genetic liabilities compared to lines with intact MMEJ, RPE1 in our case. [(1) van Overbeek, M. et al. DNA Repair Profiling Reveals Nonrandom Outcomes at Cas9-Mediated Breaks. Mol Cell 63, 633–646 (2016). (2) Tan, E., Li, Y., Velasco‐Herrera, M. D. C., Yusa, K. and Bradley, A. Off‐target assessment of CRISPR‐Cas9 guiding RNAs in human iPS and mouse ES cells. Genesis 53, 225–236 (2015)].

To further address this question of the role of ORC2, we performed a new G1 synchronization experiment with ORC2 depletion and the result is shown in new Figure 4—figure supplement 3. While arresting the cells in G1, we harvested two time points after auxin treatment. The nuclear volume remained unchanged. CENP-C staining was also unchanged (data not shown). We suggest that this abnormal heterochromatin phenotype is due to cells lacking ORC2 during S phase that affects sister chromatids cohesion loading (Zheng et al., 2018), therefore, the centromeric region showed decompaction in late S/G2 phase. We have also provided new evidence of heterochromatin decondensation after 48 hr of ORC2 depletion in Figure 4—figure supplement 2.

It is hard to tell whether aberrant mitosis is an indirect effect of incomplete DNA replication, checkpoint activation or due to a direct effect of ORC loss. Because most cells arrest at G1, late S, or G2 phase with checkpoint activation, only a very small percentage of cells enter mitosis to exhibit the abnormal phenotype. ORC2 localizes at the kinetochore during M phase, and we suggest that those mitotic cells lacking ORC2 cannot properly align and segregate sister chromatids during M phase. This could be due to direct outcome of ORC2 removal or that ORC2 is essential for recruiting other factors before M phase. In Figure 5 we’ve shown that cells can progress normally during first cell cycle upon release from double thymidine block, which indicates loss of ORC2 during mitosis doesn’t affect cell division. Nonetheless, ORC2 could still be part of an essential components for recruiting or loading important factors for chromosome congression, and this action can be traced back to G1 or S phase, which will then be hard to discriminate from its replication function.

We have added new discussion about the loss of ORC1 in the cells and the possibilities of how they can have the limited proliferation. We have also added a discussion of a more recent paper by Shibata and Dutta, 2020, on the reported loss of ORC2 and ORC5 in human cell lines.